# Interconnections between Inflammageing and Immunosenescence during Ageing

**DOI:** 10.3390/cells11030359

**Published:** 2022-01-21

**Authors:** Thibault Teissier, Eric Boulanger, Lynne S. Cox

**Affiliations:** 1Department of Biochemistry, University of Oxford, South Parks Road, Oxford OX1 3QU, UK; thibault.teissier@bioch.ox.ac.uk; 2Univ. Lille, Inserm, CHU Lille, Institut Pasteur de Lille, U1167—RID-AGE—Facteurs de Risque et Déterminants Moléculaires des Maladies Liées au Vieillissement, F-59000 Lille, France; eric.boulanger@univ-lille.fr

**Keywords:** ageing, inflammation, cytokines, inflammageing, inflammageing, immunosenescence, immunosurveillance, senescence, SASP

## Abstract

Acute inflammation is a physiological response to injury or infection, with a cascade of steps that ultimately lead to the recruitment of immune cells to clear invading pathogens and heal wounds. However, chronic inflammation arising from the continued presence of the initial trigger, or the dysfunction of signalling and/or effector pathways, is harmful to health. While successful ageing in older adults, including centenarians, is associated with low levels of inflammation, elevated inflammation increases the risk of poor health and death. Hence inflammation has been described as one of seven pillars of ageing. Age-associated sterile, chronic, and low-grade inflammation is commonly termed inflammageing—it is not simply a consequence of increasing chronological age, but is also a marker of biological ageing, multimorbidity, and mortality risk. While inflammageing was initially thought to be caused by “continuous antigenic load and stress”, reports from the last two decades describe a much more complex phenomenon also involving cellular senescence and the ageing of the immune system. In this review, we explore some of the main sources and consequences of inflammageing in the context of immunosenescence and highlight potential interventions. In particular, we assess the contribution of cellular senescence to age-associated inflammation, identify patterns of pro- and anti-inflammatory markers characteristic of inflammageing, describe alterations in the ageing immune system that lead to elevated inflammation, and finally assess the ways that diet, exercise, and pharmacological interventions can reduce inflammageing and thus, improve later life health.

## 1. Introduction

Inflammageing describes a sterile, non-resolving, low-grade, and chronic inflammation that progressively increases with age [1]. Importantly, inflammageing is not simply a marker of age but may also contribute to diminishing health, age-related diseases, and frailty [2,3,4]. However, the underlying causes of inflammageing are still somewhat unclear, with several plausible models postulated, including accumulation of cellular debris. When initially proposed, the theory of “garb-aging” [5] suggested that over time, macromolecules within cells become damaged. When coupled with progressive failure of repair and autophagy, such accumulated macromolecular damage leads to increasing levels of cellular ‘garbage’, which can trigger inflammation via innate immune signalling [1]. This theory is fully compatible with the subsequently identified hallmarks of ageing including loss of proteostasis, genomic instability (including decreased repair capacity), and altered cell signalling [6]. Although garbage accumulation may play a significant role in local and systemic inflammation, it is unlikely to be the sole driver of inflammageing. While inflammation instigated by the immune system is a critically important defence against infection, surprisingly little is known about whether and how ageing of the immune system contributes to inflammageing, and to what extent the senescence of innate and adaptive immune cells (immunosenescence) drives the chronic inflammation associated with later life. Herein, we discuss how age-related cell senescence can give rise to significant pro-inflammatory signalling and highlight the complex interplay between general cell senescence, immunosenescence, and inflammation in driving poor health in later life.

### 1.1. Senescent Cells Contribute to a State of Chronic Low-Grade Inflammation

A variety of cellular stresses can lead both proliferating and long-lived cells to enter a state of cell cycle arrest with accompanying phenotypic and functional alterations known as cell senescence [7,8,9,10,11,12,13,14]. Many of these changes are dependent upon constitutive signalling through mTOR (mechanistic target of rapamycin) [15,16,17,18], which promotes protein synthesis and an increase in biomass with resulting hypertrophy; senescent cells are also resistant to apoptosis. Senescent cells further show local DNA methylation changes [19,20] and global chromatin rearrangements [21], leading to altered gene expression patterns and the secretion of a panoply of chemokines, cytokines, and tissue remodelling enzymes. This secretome, termed the ‘senescence-associated secretory phenotype’ or SASP [22,23], is thought to have evolved to signal to the immune system for clearance [24,25] by immune cell subtypes including NK cells, macrophages, and T cells [24,26,27,28,29]. In a physiological context, senescence plays an important role in tissue remodelling during development, regeneration, and wound healing [30,31,32,33,34,35], as well as serving an important tumour suppressor role [36]. However, age-related failure of immunosurveillance, coupled with immune evasion by senescent cells (through SASP-dependent cleavage of intrinsic surface markers), leads to increasing numbers of senescent cells with age [24,37,38]. Importantly, the loss of immunosurveillance in later life is not the only mechanism leading to the accumulation of senescent cells in tissues, since senescent immune cells can also directly drive senescence in trans. This has been most clearly demonstrated in a mouse model bearing immune-specific mutation of the DNA repair factor *ercc1*, wherein the damage-driven accelerated ageing of immune cells led both to the premature accumulation of senescent cells in multiple organs and reduced lifespan [39].

The accumulation of senescent cells is likely to contribute to inflammageing, since the SASP includes pro-inflammatory cytokines (e.g., IL-6, IL-1, HMGB1, S100); chemokines (e.g., IL-8, MCP-1); soluble receptors (e.g., sTNFRs); metalloproteases (e.g., collagenase); certain protease inhibitors, e.g., SERPINs; and growth factors [23,40,41]. It is important to note that senescence is a broad term encompassing a state of terminal differentiation that depends both on the cell type and the senescence inducer and, concomitantly, that the composition of the SASP also varies significantly between cells of different lineages and according to the specific senescence inducer. However, there are a number of shared features, including SERPINs, GDF15, and STC1, that are secreted not only by lab-cultured senescent cells but that are also detected in the plasma of ageing humans [42]. The central signalling pathway for SASP generation is also likely to be shared between different types of senescent cells, converging on the transcription factor NF-κΒ, a major regulator of inflammation in immune cells that also plays a critical role in the onset of the SASP [43,44,45]. Generation of the SASP is dynamic with the composition shifting significantly over time, as shown by time-resolved proteomics analyses [46,47].

The SASP is both autocrine and paracrine in nature, triggering secondary senescence in neighbouring cells [48,49,50]. For example, the transplantation of senescent cells into the knee joint caused an osteoarthritis-like phenotype in mice, suggestive of accelerated tissue ageing [49,50], while the local clearance of senescent cells diminished osteoarthritis in the mice [51]. While such studies examined senescence in a specific compartment (the knee joint), the impact of the SASP is likely to be systemic, since local injection of senescent cells into the flank of mice led to accelerated ageing in distant tissues [52]. It has been postulated that circulating senescence-derived extracellular vesicles (EVs or exosomes) that carry SASP factors may be responsible for the development of systemic age-related diseases, even in response to initially localised senescence [50,53].

A causative relationship between the presence of senescent cells and elevated inflammation has been demonstrated both in vitro [54] and in vivo [52], reinforcing the idea of a role for senescence in inflammageing. Chronic inflammation can also induce telomere dysfunction and promote senescence, suggesting that inflammageing and senescence are mutually reinforcing [55]. Mechanistically, this reinforcement may be mediated, at least in part, through inflammatory signalling initiated by RAGE (receptor for advanced glycation end-products). Notably, adipose tissue is pro-inflammatory, and obesity and diseases such as diabetes that are associated with early onset of ageing pathologies involve the activation of RAGE signalling, which contributes to inflammation [56].

RAGE is a multiligand receptor, initially described as a receptor for advanced glycation end-products (AGEs) but which is now well-described to interact with multiple ligands, including a number of pathogen-associated molecular patterns (PAMPS) and damage-associated molecular patterns (DAMPS). RAGE signalling promotes a self-sustaining pro-inflammatory cascade, similar to that observed with other pattern recognition receptors such as TLRs [57]. RAGE expression has also been linked to physiological ageing of the kidney and blood vessels in mice, possibly by enhancing sustained low-grade inflammation and mTOR signalling [58], although to date there is no report regarding the influence of RAGE expression on lifespan. Interestingly, HMGB1, a major nuclear protein released from senescent cells [59], is a component of the SASP likely to promoting inflammageing through very high binding affinity to RAGE [57]. Not only does the SASP itself contains a wide variety of inflammatory cytokines and chemokines, but other SASP factors further amplify inflammatory signals through RAGE, NF-κB, and inflammasome stimulation; the SASP is therefore a strong candidate for driving age-related increases in inflammatory markers and is likely to contribute significantly to inflammageing.

Additional triggers of senescence include viral infection [60,61,62,63] and bacterial infection [64], leading to prolonged and harmful inflammation. In the context of an already elevated inflammatory state due to inflammageing, any additional infection-associated inflammation may cross a tolerable threshold, leading to significant tissue damage, disease, and even death. It is of particular note that COVID-19 deaths are highest in adults over the age of 65 and those with pre-existing inflammatory conditions such as diabetes and obesity [65,66,67], suggesting that viral infection exacerbates an already high inflammatory state as a result of elevated senescent cell burden with increasing age [62,63,68].

Chronic non-infectious diseases associated with ageing, as well as infectious pathogens and age-related inappropriate responses to infection, are therefore all likely to contribute to chronic inflammation, at least in part through cellular senescence; this vicious cycle means that inflammation worsens with age and more so with accompanying disease (Figure 1).

### 1.2. Inflammageing Depends on a Complex Interplay of Pro- and Anti-Inflammatory Mediators

Chronic sterile inflammation (i.e., inflammageing) increases with increasing chronological age in the majority of the human population. The overall inflammatory status is dependent not only on increases in individual markers of inflammation [69,70] but also on complex interactions between the various mediators of inflammation. Hence, attempts to assess inflammatory status need to take into account the association between pro- and anti-inflammatory markers and other parameters of ageing such as frailty, comorbidities, and lifespan, to determine whether inflammatory changes arise from increasing age per se, or from other confounding factors such as underlying age-related diseases.

#### 1.2.1. Elevation of Clusters of Pro-Inflammatory Markers with Age

Studies of circulating inflammation modulators have identified several that increase with age and may participate in the overall inflammatory burden. Individual factors include IL-8, IL-15, soluble glycoprotein 130 (SGP130, involved in IL-6 signalling), soluble cluster of differentiation 30 (sCD30), and monocyte chemoattractant protein-1 (MCP-1), all of which increase with age in peripheral blood mononuclear cells (PBMCs) or serum [71,72,73,74,75]. Of the common clinical serum biomarkers used to assess inflammation, C-reactive protein (CRP) and IL-6 are by far the most closely linked in pro-inflammatory clusters [76,77,78,79] (see Figure 2 and Table 1). They correlate well with age (especially IL-6) [75,80], poor physical performance [77,81,82], and increased death rate in longitudinal studies [79,83]. The strong association between IL-6 and CRP is not surprising, as IL-6 is known to induce the production of acute phase proteins such as CRP [69,84]. Interestingly, IL-6 on its own can drive cell senescence [85], leading to further inflammatory signalling through the SASP; hence high local levels of pro-inflammatory cytokines are self-reinforcing, potentially preventing resolution of inflammation in the context of ageing. Plasminogen activator inhibitor-1 (PAI-1, also known as SERPIN1) belongs to the same pro-inflammatory cluster as IL-6 and CRP (Table 1). Levels of PAI-1 correlate with other biomarkers of ageing, including reduced walking speed and lower grip strength [77]. While it has been suggested as a marker of frailty [86], the physiological role of PAI-1, at least in cardiac function, has been debated [87,88]. Consistent with a detrimental role of PAI-1 in ageing, the inhibition or deletion of PAI-1 limited senescence in models of accelerated ageing, both in vitro and in vivo [89,90,91].

Higher levels of IL-6 and CRP are usually interpreted as indicating a pro-inflammatory state and in older adults are used as a readout of inflammageing, but this association is not linear with age. CRP, routinely used as the best biomarker of inflammation in the clinic, is also identified as a good target for frailty diagnosis [92]. The effect of a high inflammatory score on mortality was observed to be greatest at age 65–70, while the link between inflammation and age diminished in subjects ≥80 years old, in studies of two independent cohorts [83]. Furthermore, increased plasma levels of both IL-6 and CRP in adults over 90 years of age were not associated with increased mortality, when fully adjusted for covariates [93]. It is possible that the relative importance of inflammation might decrease at very old ages when other biological processes, such as stem cell exhaustion, kidney dysfunction, cardiovascular disease, cancer, or dementia, may play a greater role in age-related morbidity and mortality. Alternatively, people who have survived beyond 90 years may represent a relatively unusual subset of individuals who are physiologically more robust and less prone to age-related decline—indeed, studies of supercentenarians and their offspring strongly support this view of centenarian genetic exceptionalism [94,95,96,97].

**Table 1 cells-11-00359-t001:** Examples of pro-inflammatory factors identified in major cohort studies. Note this is an indicative and not exhaustive list, covering findings from a subset of large population cohorts, and does not include all known inflammatory factors (NI = not investigated in the study cited).

IL-1β	IL-6	CRP	SAA	ICAM	PAI-1	Fibrinogen	Population Characteristics	Data Processing	Reference
No cluster reported	PCA1—Correlated with age, mortality, morbidity, and age-related diseases		NI	NI	NI	NI	1010 participants (428 men, 582 women), aged 21–96 y.o.	Cluster analysis	[75]
Absence of correlation with age	Positively correlated with age, mortality, morbidity, and some age-related disease	Positively correlated with age, mortality, morbidity, and some age-related disease					General population—InCHIANTI cohort	Individual associations	
Non-predictive of 5-year mortality	Significant independent predictor of 5- and 10-year mortality	Significant independent predictor of 5- and 10-year mortality	NI	NI	NI	NI	7043 participants (2995 men, 4048 women), aged 65–102 y.o.	Fully adjusted correlation with 10-year mortality	[83]
	Best predictor of 10-year mortality in combination with sTNFRI						General population—includes CHS and InCHIANTI cohorts		
IL-1/TGF—No adverse outcome reported	“Up”-regulation—Positively correlated with worsened mobility and frailty risk		NI	NI	NI	NI	967 participants (416 men, 551 women), aged 65 y.o. and older	Cluster analysis	[76]
							General population—InCHIANTI cohort		
NI	CRP related—Negatively correlated with grip strength; Positively correlated with 400-m walk time		NI	NI	CRP related—Negatively correlated with grip strength; Positively correlated with 400-m walk time	NI	1269 participants, aged 70–79 y.o.	Cluster analysis	[77]
	Highest *R*^2^ of the CRP-related component for knee and grip strength	Highest *R*^2^ of the CRP-related component for walking speed and 400-m walk time			Highest *R*^2^ of the CRP-related component for the HPPB score		General non-frail population	Individual associations	
NI	Positively associated with 4-year mortality		NI	NI	NI	NI	285 participants (67 men, 218 women), aged 90 y.o. and older with a 4-year follow-up	Correlation with 4-year mortality	[93]
	Removed association with mortality	Predictor of 4-year mortality alone or in combination with IL-1RA					General population	Fully adjusted	
NI	NI	Systemic inflammation	NI	Local inflammation-endothelial dysfunction	NI	Systemic inflammation	320 participants (236 men, 84 women), aged 58–74 y.o. with a 1-year follow-up	Cluster analysis	[78]
		Independent predictor of 1-year recurrent coronary events		No association with 1-year recurrent adverse cardiac events		No association with 1-year recurrent adverse cardiac events	Acute coronary syndrome patients	Individual associations	
Absence of correlation with age	Positively correlated with age					Positively correlated with age	1327 participants (586 men, 741 women), aged 20–102 y.o.	Correlation with age	[80]
NR	Greatly reduced the size of the regression coefficient for age	Removed the effect of age in men only				Removed the effect of age	General population—Includes InCHIANTI cohort	Adjustment for cardiovascular risk factors and morbidities	
NI	Proinflammation—Strong association with 4-year death rate			No cluster reported but is individually positively correlated with TNF-alpha, CRP, IL-6 and SAA	NI	NI	580 women aged 31–85 y.o. with a 4.7-year follow-up	Cluster analysis	[79]
							Clinically referred for coronary angiography		
No significant association with physical performance	Negatively correlated with global physical performance and hand-grip strength	Negatively correlated with global physical performance and hand-grip strength	NI	NI	NI	NI	1020 participants (483 men, 537 women), aged 65–102 y.o.	Correlation with physical performance	[81]

In defining a role for inflammatory mediators in age-related morbidity and mortality, it is important to remember that IL-6 can also act in an anti-inflammatory capacity. Thus, an increase in IL-6 does not necessarily mean that there is an increase in pro-inflammatory processes. It has been suggested that levels of both IL-6 and CRP can be moderately increased without being associated with elevated inflammation [98]. Indeed, studies on muscle physiology have shown that on exercise, muscles release into the circulation hormone-like molecules with anti-inflammatory properties called myokines, which include IL-6 [99]. In this context, IL-6 production is independent of NF-κB or TNF-α activation [100] and exerts an anti-inflammatory effect by stimulating the production of IL-1RA and IL-10 [99,101], while the production of TNF-α by macrophages is diminished [102]. However, increased NF-κB signalling and downregulation of NF-κB attenuators is reported in ageing muscle [103], suggestive of chronic inflammatory signalling. Therefore, a high serum level of IL-6 in healthy young people or in frail older adults may indicate opposite situations of anti- and pro-inflammatory activity, so that IL-6 concentration per se is not informative of inflammatory status without additional contextual information including age, disease, and exercise status.

Other inflammatory mediators also appear to have both pro- and anti-inflammatory activity in different biological contexts. For instance, IL-18 is usually considered to be a pro-inflammatory factor [104,105], but its role in inflammation is not one-sided, and it clusters between pro- and anti-inflammation groups [78,79] (Figure 2, Table 2). IL-18 acts as an anti-inflammatory cytokine in early-stage gut inflammation, with pro-resolving properties, while it appears to be pro-inflammatory at later stages [106,107]. Consistent with pro-inflammatory action, IL-18 levels positively correlate with age [75,80] and are a significant independent predictor of 5-year mortality in a longitudinal study [83]. Tumour necrosis factor alpha (TNF-α) is often considered as a pro-inflammatory marker similar to other ‘strong’ markers of inflammageing such as IL-6. However, it has repeatedly been included in different clusters than IL-6 [76,77,79], indicating a very different pattern of induction. Moreover, while TNFα is associated with frailty [82], its levels do not correlate with age [75,80].

#### 1.2.2. Do Anti-Inflammatory Factors Contribute to Inflammageing?

Anti-inflammatory factors, by their nature, would be expected to reduce inflammation including inflammageing. This is indeed the case with the potent anti-inflammatory cytokine IL-37 [108,109], which has an unequivocal effect in damping down harmful inflammation. For example, transgenic mice expressing human IL-37 showed little age-related increase in pro-inflammatory cytokines and maintained B cell progenitor function with age [110]. Moreover, IL-37 administration significantly reduced inflammation, restored T cell function, and ameliorated vascular, metabolic, and motor function in old mice [111,112,113]. Similarly, levels of the anti-inflammatory cytokine IL-10 [114] negatively correlate with age [75], and high IL-10 was not associated with any detrimental outcomes in physical performance [81]. In fact, IL-10 was included in an anti-inflammation cluster together with HDL cholesterol and is associated with good prognosis in acute coronary syndrome patients (~67 years old) [78].

Counterintuitively, however, a number of factors classified as anti-inflammatory appear to correlate with poor physiological function when they increase with age. For instance, soluble TNF receptor I (sTNFR-I) and II act as inhibitors of TNF-α pro-inflammatory signalling by preventing its interaction with membrane TNFR. Hence they would be anticipated to be anti-inflammatory, yet they have a strong positive correlation with increasing age [75] and are associated with a lower physical performance [77]. An index that combined IL-6 and sTNFR-I was the best predictor of 10-year mortality in a longitudinal study of two large cohorts of older adults [83], suggesting that sTNFR-I acts with IL-6 to promote inflammation.

The interleukin-1 receptor antagonist (IL-1RA) blocks IL-1 signalling and has been described as anti-inflammatory. However, levels of IL-1RA positively correlate with age [75,80], and it clusters with IL-6 and CRP [76,81]. In fact, IL-1RA is a significant predictor of mortality in older adults independent of inflammatory exposure or genotype [83,93]. Two other soluble interleukin receptors, IL-2 soluble receptor (IL-2sR) and IL-6sR, have been included in the same cluster as TNF-α and sTNFRs and are associated with poor physical performance [77]. However, in another study, IL-6sR was not associated with negative performance [81] and was not correlated with age in men [75], though this may not be the case in women [80]. Significant sex-specific differences in inflammatory cytokine levels in response to infection have been described in patients with COVID-19 [115], highlighting the importance of accounting for biological sex when interpreting cohort findings on inflammatory markers and inflammageing. The influence of biological sex on inflammatory responses may in part be due to selective X inactivation in women (the single X chromosome in men is constitutively active). De-repression of previously silenced immune loci on the X chromosome with age may account for the increase in pro-inflammatory autoimmune conditions and X-skewing in older women [116].

The association between multiple anti-inflammatory factors and age-related disease does not necessarily mean that these molecules are detrimental and cause inflammageing. Instead, their elevation may reflect an adaptive physiological response to pro-inflammatory stimuli, to try to damp down the inflammation. Where high levels of anti-inflammatory factors are associated with poor outcomes, it could be that the adaptation failed to reduce inflammation below a damaging threshold. Thus, it has been suggested that the balance between IL-1RA and IL-1 may be more important than IL-1RA levels alone in determining inflammatory outcomes [93,117,118]. Despite this, some studies have shown that plasma IL-1β was not associated with poor outcomes [76,81] and did not correlate with age [75,80], making IL-1RA a better marker of inflammation than IL-1β. Transforming growth factor β (TGF-β) [114] shows similar regulation to IL-1β in older adults [76] and levels did not correlate with age [75,80].

Despite the lack of consistency in the markers analysed between different studies, a conserved pattern can be identified in which a few inflammatory mediators appear key to an ageing signature of inflammation i.e., inflammageing. Based on this, we suggest that the determination of inflammageing needs more than simple measures of the level of individual cytokines and that instead a full range of inflammatory mediators (pro- and anti-inflammatory) should be analysed, as well as the context in which high levels are seen, such as tissue, disease state, biological sex, and chronological age. Drawing together the studies discussed above, we identify a possible ‘ageing signature’ of inflammatory mediators (see Figure 2 and Table 1).

#### 1.2.3. Inflammation Can Drive Morbidity in Older Adults—Thromboembolitic Events as an Example

While acute inflammation is an evolved protective response, chronic inflammation, as seen in inflammageing, has detrimental consequences across multiple tissue and organ systems, leading to age-related disease and increasing the risk of death. This is exemplified by the tight link between inflammation in older age and changes in blood viscosity and elevated clotting risk [119], with direct mechanistic links suggesting that inflammation may trigger clotting [120]. Venous thrombosis is rare in young adults (1 in 10,000 per year) rising to an incidence of 1 in 100 in the elderly; circulating clots can lead to heart attack, pulmonary embolism, and/or ischaemic stroke, with many incidents leading either to long-term disability or death. While major risk factors include malignancy and other comorbidities [121], age-related inflammation may also pay a part. In one study, fibrinogen clustered with CRP in the “systemic inflammation” group [78]; it has also been shown to correlate with age [80]. High serum levels of coagulation factors such as D-dimer (a fibrin degradation product) are associated with functional decline and increased mortality in older persons [122,123]. This is relevant not only in ‘normal’ ageing but also in infectious diseases in the elderly—increased plasma viscosity resulting from high fibrin levels with extremely elevated D-dimer measurements is associated with markedly increased inflammation and risk of cardiovascular disease and thrombotic events in older adults with COVID-19 [65,124]. As well as fibrinogen, other coagulation factors such as von Willebrand factor; prothrombin activation peptides; factor VII, VIII, IX, or X; and thrombin-antithrombin complexes are modulated with age and associated with a state of hypercoagulability in elderly subjects [125,126,127]. It is possible that inflammageing and hypercoagulability share a common pathway with senescence that involves insulin-like growth factor binding protein-5 (IGFBP-5) [128].

## 2. Immunosenescence Affects Many Components of the Immune System

Immunosenescence, the ageing of the immune system, is closely related to inflammageing. With time, the immune system becomes less responsive to pathogens, and vaccine efficacy is also often reduced with age [129]. The ability to initiate acute inflammation has survival benefit, and a lack of responsiveness to acute inflammation is associated with higher blood CRP levels and increased mortality risk [130]. This failure results in increased susceptibility to various infectious diseases, including common infections that are relatively harmless in healthy adults (for example, influenza infections do not markedly differ between old and young people, but deaths from influenza are greatly elevated in the elderly [131,132,133]). Immunosenescence and the failure of protective innate responses may also contribute to the high mortality of older adults on SARS-CoV2 infection [134]. Below, we discuss dysfunction of various components of the immune system, including barriers, innate immune cells, and the adaptive immune system (Figure 3) in terms of ageing and senescence.

The epithelium forms the primary barrier against pathogen infection. Secretions such as mucus, saliva, or tears help in the removal of pathogens via mechanical and enzymatic actions [135,136,137]. Age-associated decline in the secretion of mucous and tears through changes to aquaporins [138], as well as decreased cilia function [139], may lead to increased pathogen dwell times in target tissues, increasing the risk of infection. The loss of specialised function through senescence-associated changes in chromatin conformation (hence altered gene expression) may be responsible for the observed reduction in barrier efficacy. It is of note here that older adults are at much higher risk of community-acquired pneumonia than younger adults, potentially through diminished capacity to capture and expel pathogens from the respiratory epithelium [140,141]. Epithelial barriers gradually lose integrity with age, directly through the senescence of the epithelial cells themselves, and possibly also through disorganised tight junction proteins leading to gaps between cells through which pathogens can pass. This is particularly relevant in the gastrointestinal (GI) tract, where gut leakiness can result in the exposure of tissues to bacteria bearing the potent PAMP, lipopolysaccharide (LPS) [142,143]. In addition, changes in the integrity of the blood–brain barrier may permit the entry either of immune cells [144,145,146,147] or pathogens (e.g., gum disease bacteria *Porphyromonas gingivalis* have been mooted as candidate pathogens driving central nervous system (CNS) inflammation and dementia [148,149,150]).

Innate immunity is the oldest and most conserved aspect of immunity in animals, although its complexity varies greatly between species [151]. The innate response represents a rapid but relatively non-specific reaction to infection or injury. Innate immune signalling is first triggered by the binding of conserved PAMPs or DAMPs [152,153] to pattern-recognition receptors (PRRs) of ‘professional’ innate immune cells [152,153]. Many other cell types, including epithelial cells and fibroblasts, also possess PRRs and can initiate innate immune responses.

The PRRs fall in to several distinct classes including: Toll-like receptors (TLRs), nucleotide oligomerisation domain (NOD)-like receptors (NLRs), retinoic acid-inducible gene (RIG)-I-like receptors (RLRs), and C-type lectin receptors (CLRs), as well as cytosolic DNA sensors, such as cyclic GMP-AMP synthase (cGAS) or absent in melanoma 2 (AIM2) [154,155,156,157,158]. In addition, RAGE may act as a PRR with a specific and important role in inflammageing [57,159]. Signalling through these pathways activates a cascade of pro-inflammatory responses mediated through key nodes, including the inflammasome and transcription factor NFκB (see below). The accumulation of DAMPs and PAMPs with age will therefore induce a systemic pro-inflammatory response. Notably, senescent cells produce a number of DAMPS, including HMGB1 and cytosolic DNA from damaged mitochondria DNA, cytoplasmic chromatin fragments [21,160], and LINE1 transposon reactivation [161], suggesting that the accumulation of senescent cells with age may stimulate an ongoing non-resolved innate inflammatory response. Indeed, the lack of clearance or the excess production of molecular “garbage” has led to the “garb-aging” theory [5].

Following the breaching of epithelial barriers by a pathogen, in mammals, the complement system is activated and can provoke the lysis of targeted pathogens. Complement activation is accompanied by cellular responses involving tissue-resident immature dendritic cells (DCs), mast cells, granulocytes (encompassing basophils, eosinophils, and neutrophils), natural killer cells (NKs), and macrophages [162]. Some cells have a role in both innate and adaptive immunity, including cytotoxic γδ T lymphocytes and NKT cells [162] (see Figure 4). The contribution of specific innate immune cell types and their progenitors to inflammageing is discussed further below.

A failure of any component of these early innate defences with age is likely to both increase susceptibility to infectious disease, as well as elevate the risk of chronic inflammation and subsequent morbidity. Furthermore, the failure of the resolution of the initial inflammatory trigger leads to chronic inflammation with tissue injury (reviewed in [163]).

## 3. Dysregulation of Innate Immunity on Ageing

### 3.1. Ageing of Haematopoietic Stem Cells Leads to Myeloid Skewing

All professional innate and adaptive immune cells originally arise from haematopoietic stem cells (HSCs) in the bone marrow (Figure 4). With increasing age, HSCs become less quiescent [164] and even though they increase in number [165,166], they exhibit a functional decline in regenerative potential [165,167,168], possibly through telomere-driven replicative senescence. The capacity of aged transplanted HSCs to home to the bone marrow is also greatly decreased compared with young HSCs [167,169], suggesting intrinsic impairment of aged HSCs. By contrast, the mobilisation of hematopoietic stem and progenitor cells from the bone marrow to peripheral blood was greater in old versus young mice, possibly as a consequence of reduced adhesion to stroma [170]. These defects might arise from the accumulation of DNA damage, epigenetic changes, increased ROS production, altered proteostasis (all features of cell senescence), and/or changes in cell polarisation [171,172,173].

The impact of extrinsic factors on HSCs is still not well understood [171], although increased inflammation may negatively impact on HSC function [165,174,175]. Age-related accumulation of senescent cells in the bone marrow (including mesenchymal stem cells, as well as bone macrophages and long-lived osteocytes) may contribute, through the SASP, to a pro-inflammatory microenvironment that favours the differentiation of HSCs towards myeloid progenitors [176] at the expense of lymphoid progenitors (adaptive immune B, T cells and NK cells) [164,165] (see Figure 3). Chronic exposure of HSCs to other inflammation-promoting molecules may be important in driving this bias, as elevated chemokine (C-C motif) ligand 5 (CCL5) is thought to promote myeloid skewing [174,177]. Inflammation can arise through many routes, but one driver of particular interest in ageing is the presence of bacterial LPS in tissues through the loss of barrier function (see above); whether this has a direct impact on HSC differentiation is not yet clear. Overall, myeloid-skewing predisposes immune responses towards innate processes, which are generally pro-inflammatory, and is likely to reduce adaptive immune clearance of infection, leading to prolonged inflammation.

### 3.2. Ageing in Innate Immune Effector Cells

#### 3.2.1. Dendritic Cells

Dendritic cells are major antigen-presenting cells (APCs) that display processed antigens in conjunction with MHCII (major histocompatibility complex II) to activate helper T cells [178,179]. In older adults, DCs show diminished function, including impaired migration and phagocytosis, increased production of TNF-α and IL-6, and decreased production of IL-10 upon stimulation [180,181,182]. (Note, however, that the assessment of cytokine release may depend on whether or not DCs are expanded in vitro prior to activation [183]). Decreased phagocytosis by ageing DCs leads to decreased antigen presentation and hence, reduced T and B cell priming [183,184,185]. This is relevant both in the context of age-associated autoimmunity (since self-tolerance relies on elimination of self-reactive T cells) [186] and in age-related susceptibility to infectious diseases. Whether the numbers of plasmacytoid and myeloid DC are altered with ageing is less clear, as some reports find a decrease in both cell types [187], others in only one cell type [188,189], while still others report no decrease at all [180]. Overall, however, it is clear that DC function decreases with age and that this contributes to poor immunity in the elderly.

#### 3.2.2. Granulocytes

Neutrophils serve important roles in direct pathogen killing and migrate rapidly to sites of infection in response to inflammatory cytokines. The numbers of neutrophils appears to stay constant over the life course [190,191]. While their chemokinesis (i.e., speed of movement) appears to be unaltered with ageing, importantly, their phagocytic capacity and directional movement towards a chemical stimulus (i.e., chemotaxis) both diminish with age [191,192,193]. This poor directionality results in additional damage to tissues through the action of tissue-degrading neutrophil elastases as well as local neutrophil-mediated inflammation [194,195,196,197]. Recent data showed that neutrophils in aged mice had a high frequency of reverse transendothelial migration, followed by dissemination to the lung, contributing to remote organ damage and possibly systemic inflammation [198]. Neutrophils produce bursts of reactive oxygen species (ROS), which can directly kill pathogens, but this intense acute ROS production is significantly lower in neutrophils from older adults compared with those from young donors [199,200,201]. However, neutrophils from older adults instead show higher basal ROS levels, which could indirectly contribute to inflammageing and reduce effective response to pathogens [200,202]. Moreover, the propensity of neutrophils to extrude their genomic DNA to trap pathogens in a process known as NETosis—another pathogen killing strategy—also shows an age-related decline [203]. Hence, age-related alterations in neutrophil function delay pathogen clearance and wound healing [204] and increase local inflammation.

Signalling pathways by which neutrophils respond to cytokines appear to be altered with ageing. For example, responses to granulocyte–macrophage colony-stimulating factor (GM-CSF) diminish with age [205,206]. Whether aged neutrophils exist in a state of hyperactivation is moot—reports of decreased expression of CD50 and CD62L (L-selectin), increased spontaneous ROS production, and the upregulation of pro-inflammatory pathways all suggest hyperactivity of old neutrophils [200,207,208], in contrast with other studies reporting little difference in hyperactivity markers between neutrophils from young and old donors [199].

Since aged neutrophils can still sense inflammatory cytokines but fail to deal appropriately with pathogens, it is likely they are defective in signal transduction pathways downstream of ligand binding to receptor [203,209]. Consistent with this, the mobilisation of the second messenger calcium [191,210], as well as the polymerisation of actin, are reduced in neutrophils from older adults [211,212]. Cell motility and phagocytosis both require a dynamic actin cytoskeleton, so signalling via the small G-protein Cdc42 to actin is likely to be altered in old neutrophils. Indeed, the treatment of aged neutrophils with statins, which act on Cdc42 (in addition to their role in cholesterol regulation), improves neutrophil chemotaxis and reduces NETosis [141,203]. Cdc42 inhibition may also be beneficial in reducing inflammation from other sources, such as senescent vascular endothelium [213].

Compared with the wealth of data on neutrophils, the effect of ageing on eosinophils and basophils has been much less studied [214]. The data for basophils appear contradictory according to the species studied. For example, in mice, spleen basophils, but not bone marrow basophils, increase in number during ageing [215], whereas in macaques, daily basophil production decreases with age, but their half-life remains stable [216]. At least one study has suggested a reduction in basophil numbers in the blood from healthy older humans [217], but basophil degranulation in response to allergens does not appear to change with age [218]. By contrast, eosinophil degranulation upon stimulation and possibly superoxide anion production were negatively affected by age, but adhesion and chemotaxis were conserved [219]. Some reports indicated a deficit of eosinophils at infection sites with age, suggesting a potential alteration in chemokinesis [220,221]. It is important that this gap in knowledge is addressed—basophils may be important in Th2-mediated responses to allergens and multicellular pathogens [222], as well as regulating autoimmunity, yet whether diminished basophil function is involved in age-related autoimmunity is not currently known.

#### 3.2.3. Natural Killer Cells

Natural killer (NK) cells bridge the innate and adaptive immune system—they derive from lymphoid lineages but act as innate effector cells (Figure 4), directly killing infected host cells as well as tumour cells expressing neoantigens. There is some conflict in the literature concerning how NKs change with age that may reflect species-specific differences—levels and inflammatory responses of classical NK T cells have been reported to increase with age in mice [223,224] though decrease in older humans [225,226,227,228,229]. When considering functional NK cells, levels of immature NKs, which express high levels of surface CD56 (CD56^bright^), diminish with age [230,231], while mature NK cells, with low CD56 (CD56^dim^), increase [230,231,232]. Even in cases wherein overall NK levels increase with age, there is likely to be reduced functionality since cytotoxicity per cell (e.g., lower perforin release) decreases [233,234,235,236,237].

NKs are activated by IL-2, which is produced by helper T cells (Th1). Interestingly, IL-2 levels decrease with age or with increasing frailty [238,239,240], which is likely to have knock-on effects on NK activation (as well as on CD8^+^ T cell function—see later). Furthermore, the response of NKs to IL-2 is diminished with age, as observed by reduced proliferation, Ca^2+^ mobilisation, and the expression of CD69 (an early activation marker) [225,241]. However, non-stimulated NK cells show elevated CD69 expression with age, suggesting higher basal activation levels but decreased responsiveness to activation signals [234]. Whether the production of immune modulators by NK cells changes with age is more contradictory, with some reports demonstrating stable TNF-α and IFN-γ induction [225,234], whereas others report age-associated decreases in the production of IFN-γ, IL-8, and macrophage inflammatory protein-1α (MIP-1α) by NKs [133]. NK cells play an important role in clearing senescent cells [24,28,37], especially senescent tumour cells (reviewed in [26]), so reduced NK activity with age may contribute to the accumulation of senescent cells.

#### 3.2.4. Macrophages

Macrophages are at the centre of both the initiation and resolution of inflammatory events. Initiation is characterised by the secretion of pro-inflammatory cytokines (IL-6, IL-1β, TNF-α), reactive oxygen species (ROS), nitric oxide, and neutrophil chemokines. Resolution involves pathogen removal, the downregulation of neutrophil chemokines, and the removal of apoptotic neutrophils (efferocytosis), which tilts equilibria towards an anti-inflammatory state [242,243]. Macrophages also produce specialised pro-resolving mediators (SPMs), a class of molecules including resolvins, maresins, protectins, and lipoxins derived from some poly-unsaturated fatty acids (such as eicosapentaenoic acid or docosahexaenoic acid) and have recently been described as potent factors able to resolve both innate and adaptive pro-inflammatory events [244,245,246,247,248]. By contrast, molecules derived from arachidonic acid (apart from lipoxins) are predominantly pro-inflammatory [248]. A correct balance of these modulators is likely to be important in ensuring the appropriate activation and resolution of inflammation [249,250,251,252], so their dysregulation could lead to inflammageing and tissue ageing. While one small study showed a profound reduction of lipoxin A_4_ levels in urine samples from elderly people [253], to date, there has been little research on the role of SPMs in human ageing per se, though they have been linked to age-related diseases [254]. Aged mice have been reported to show delayed resolution of acute inflammation [255], which may be linked to lower levels of SPMs [256,257,258]. Low SPM levels in older mice can be overcome by exogenous SPM delivery, which reduced inflammation and accelerated the resolution of peritonitis [255]. Mechanistically, it appears that decreased SPM levels may result from age-associated promoter methylation, which reduces gene expression. For example, *Elovl2*, a gene involved in SPM precursor synthesis, is progressively hypermethylated with age, while the reversal of *Elovl2* promoter hypermethylation in the retina of ageing mice rescued age-related decline in visual function [259]. It has been suggested that levels of ELOVL2 may therefore serve as a sensitive, non-tissue-specific biomarker of ageing [260,261,262].

With a half-life of several weeks [263], macrophages are much longer lived than the majority of innate immune cells, whose half-life rarely exceeds a few days in vivo [216,264,265]. Macrophages therefore survive long enough to accumulate molecular damage associated with senescence. This has the potential for significant pathological effects, since senescence-associated changes in lysosomal pH [266] may prevent macrophages from breaking down engulfed pathogens or cell debris. For example, high lysosomal pH in microglial cells, which serve as central nervous system macrophages, makes them unable to degrade Aβ amyloid peptides [267], leading to amyloid deposition and CNS inflammation associated with Alzheimer’s disease [268].

As well as serving key roles in inflammation and stress responses, macrophages can also be considered to be at the centre of inflammageing, since multiple stressors increasingly stimulate macrophages over time [1]. While overall numbers of monocytes and macrophages do not change with age [269], the most pro-inflammatory subsets of monocytes increase with age, thereby contributing to inflammageing [270], and exhibit dysregulation of pro-inflammatory and phagocytic functions [72,270,271]. Macrophage phagocytosis appears to be maintained with age, except in some tissues such as the peritoneum, suggesting that the microenvironment can influence macrophage function [272,273,274]. Macrophages from old mice show diminished cytokine release, which may be due in part to negative feedback responses to an already elevated inflammatory environment in aged mice [275,276,277,278,279,280], since a reduction in global inflammation (in an *Il-6*^−/−^ genetic background) resulted in macrophages from old mice exhibiting a stronger inflammatory response than those from younger mice [275]. This suggests that the high senescent cell burden in old age, with associated SASP and chronic inflammageing, may block the appropriate activation of professional innate immune cells following exposure to pathogens, which could account for the increased risk from infectious diseases with increasing age.

### 3.3. The Inflammasome Drives Pyroptosis and Further Inflammation

Innate responses to infection, tissue damage, or certain environmental triggers (e.g., asbestos) can lead to a specialised form of lytic cell death that is highly inflammatory, termed pyroptosis [281,282]. It is triggered through activation of the inflammasome, a cytosolic multiprotein complex that responds via Toll-like, Nod-like, and AIM-like receptors (TLRs, NLRs, and ALRs) to microbial and cellular damage (PAMPs and DAMPs) [283], as well as extracellular ATP and potassium efflux from the cell. The inflammasome forms a scaffold which activates caspase-1 (caspase-4, 5, or 11 can also be involved) [284,285], leading to the cleavage of gasdermin D and precursors such as pro-IL-1β and pro-IL-18 [154] into active IL-1β and IL-18 [282,285,286], as well as the formation of pore-like components that are responsible for membrane destruction and pyroptotic death. The inflammasome has been implicated in a number of metabolic and age-related diseases, including diabetes, autoimmunity, Parkinson’s, and Alzheimer’s disease (reviewed in [287]).

The NLR, pyrin domain containing 3 (NLRP3) inflammasome is the most studied inflammasome and is particularly important in the link between inflammation and ageing. NLRP3 has been implicated in the systemic inflammation that characterises metabolic diseases, while its deletion significantly limits this inflammatory phenotype [288,289]. In the context of ageing and inflammageing, NLRP3 is activated by age-related DAMPs, such as extracellular ATP, urate, ceramides, or palmitate. Cytokines such as TNF-α, levels of which increase with age in mice, are also able to prime the NLRP3 inflammasome [290]. Importantly, mitochondrial dysfunction can trigger NLRP3 inflammation, since damaged mitochondria release cardiolipin and oxidised mitochondrial DNA (mtDNA) that are recognised as DAMPs and promote NLRP3 priming [291,292]. Notably, NLRP3 deletion reduced inflammatory innate immune activation in the ageing central nervous system and ameliorated age-related decline in cognitive function, strongly suggesting that excess CNS inflammation can promote cognitive decline, a common feature of ageing [293]. Such effects were not confined to the CNS as metabolic function, motor function, and bone structure were also better preserved upon NLRP3 deletion [293], and age-related thymic involution and reduction in naïve T cells pools were significantly prevented [293,294]. Moreover, in a mouse model of age-related clonal haematopoiesis of indeterminate potential (CHIP), detrimental phenotypes of increased clonality and atherosclerosis, together with the overexpression of NLRP3 and IL-1β, were largely prevented by the use of an NLRP3 inhibitor [295]. Importantly, inflammasome components caspase-1 and ASC were found to increase with age in mice and in humans, including in a longitudinal study of cells from the same human donor, with susceptibility to oxidative stress being ameliorated by treatment with an inflammasome inhibitor [296]. This finding of increased ASC is critical for the propagation of inflammageing, as ASC-mediated specks can be phagocytosed from the extracellular space by macrophages, triggering lysosomal damage and further inflammation [297]. While direct studies on ageing are limited, it is becoming increasingly clear that the inflammasome is intrinsically linked to inflammageing and age-related diseases.

### 3.4. Epigenetic Changes Provide Innate Immune Cells with Memory

Although innate immunity has long been considered non-specific and to lack memory, it now appears that it is able to adapt and retain some form of memory, termed “trained immunity”, which appears to rely upon epigenetic changes [298,299,300,301] including patterns of CpG methylation, post-translational modification of histone tails (the ‘histone code’), and longer-range chromatin interactions involving looping into transcriptional activation domains [302]. Such epigenetic remodelling is thought to lead to a quicker and stronger immune response on re-exposure to the same PAMP or DAMP. Trained monocytes, key contributors to innate immunity, show increased H3K4me3 [298], which has been associated with upregulated production of pro-inflammatory cytokines [303]. It is therefore tempting to suggest that trained immunity, while beneficial in the short term, may also produce long-term effects favouring a sustained increase in inflammatory status. Indeed, innate immune microglial training increased Alzheimer disease (AD) features including neuronal loss and cognitive decline in a streptozotocin-induced AD model in mice, which was further exacerbated by peripheral LPS-induced inflammation. Notably, the inhibition of the NLRP3 inflammasome protected against microglial training in this model and reduced neurological pathologies [304].

Epigenetic information is plastic, reflecting environmental and endogenous events over the cellular and organismal lifespan (the ‘exposome’) [305,306,307], though it can also be inherited trans-generationally [308]. Major alterations in epigenetic information occur during cell senescence, including global loss of heterochromatin [309,310], reduced levels of histone trimethylation H3K9me3 [311,312] and hypomethylation at most CpG sites in DNA [160,313], with hypermethylation on a few CpGs. The methylation status of just 353 CpG sites can be used as an ageing ‘clock’ to determine both chronological and physiological age [19,20]. Such methylation patterns can be used to identify individuals at risk of poor ageing outcomes and even predict all-cause mortality [314]. By contrast, people who show ‘successful ageing’ i.e., without overt morbidity at late chronological age, such as super-centenarians and their offspring, show a lower epigenetic age than expected [95].

Innate immune cells are not exempt from these senescence- and age-related epigenetic alterations, including shifts in patterns of methylation that lead to the expression of genes that should be silenced, and the silencing of genes that should be expressed. For example, the age-related demethylation of the *TNF*-α promoter is associated with the higher expression of TNF-α in human leucocytes, monocyte-derived macrophages, and PBMCs [315,316]. Similarly, the demethylation of the CXCL10 (C-X-C motif chemokine 10) promoter in lymphocytes and neutrophils also leads to its upregulation [317]. By contrast, the age-related hypermethylation of *ATG5* and *LC3B* promoters in macrophages is associated with decreased autophagy, contributing to chronic inflammation [318,319]. Taken together, these studies suggest that epigenetic changes associated with ageing and with senescence may contribute to inflammageing through increasing pro-inflammatory gene expression and repressing anti-inflammatory factors.

## 4. Dysregulation of Adaptive Immunity during Ageing

Adaptive immunity is activated subsequent to innate immunity and when the innate response to the pathogen is insufficient to clear infection. It is a slower process, taking days or even weeks compared with the very rapid minutes to hours for the innate response but is much more specific to particular antigens. The key players are the B and T lymphocytes, which are activated upon MHCII-mediated antigen presentation by professional APCs (usually dendritic cells) through the immunological synapse, together with type I interferon and cytokine signalling. Common lymphoid progenitors (CLPs) in the bone marrow produce small lymphocytes that will become B cells if processed in the bone marrow [320] or T cells if processed in the thymus [321]. T cells carry out cell-mediated responses [322,323,324], while B cells predominantly produce antibodies in the humoral response. Lymphocytes can differentiate into multiple different types, designated by the presence of surface markers. In this discussion of immunosenescence, we consider the major classes of CD4+ helpers and CD8+ cytotoxic, with regulator, effector, and memory sub-classes, together with antibody-secreting plasma B cells and memory B cells.

### 4.1. Decreased B and T Cell Responses with Ageing

#### 4.1.1. B Cells

B cells express unique immunoglobulin B-cell receptors (BCRs) on their surface, derived through V(D)J recombination and the somatic hypermutation of the immunoglobulin Ig genes. Upon binding to antigens, presented by DCs or other professional APCs, and stimulation by T helper cells, they undergo clonal proliferation, resulting in millions of identical plasma cell clones [325,326]; following Ig class switching, they secrete an antibody specific to the target antigen [325]. Circulating Ig can neutralise pathogens displaying the cognate antigen, which can then be lysed by the complement system. Antibodies can also opsonise the pathogen to enhance phagocytosis by macrophages [327,328,329]. Some long-lived B cells will remain as quiescent memory cells after the initial response has been completed; these memory cells can rapidly respond on the subsequent presentation of the same antigen [325].

An overall decrease in naïve B cells, including pro-, pre-, and immature B cells, is seen in older adults [330]. This is a consequence both of B cell-intrinsic ageing [331,332], as well as age-related defects in the bone marrow and lymph nodes [333]. Existing memory B cells persist but further B_mem_ differentiation is impaired in older adults [334], because of the low expression of Blimp-1, a master regulator of plasma cell differentiation. An atypical subset of non-dividing B cells, termed age-associated B cells (ABC), accumulates with age [335,336]. They are able to present antigens, to secrete cytokines and antibodies, and are responsive to innate receptor stimulation but are unresponsive to BCR stimulation [335,337]. Their presence is highly dependent on inflammatory processes, notably the stimulation of TLRs [336], which favours pro-inflammatory innate immune responses [337]. Conversely, pro-inflammatory IFN-γ and IL-15 were negatively associated with B cells in the bone marrow [338], although none of these molecules has been reported to increase in the circulation with ageing. In addition, ABCs are implicated in the production of self-reactive antibodies important in the emergence of autoimmunity [336,337]. Indeed, the selective destruction of these cells in a murine model significantly diminished autoantibody production [336]. Regulatory B cells (Breg) have strong immunomodulatory properties and can resolve pro-inflammatory events mostly by producing IL-10, IL-35, or TGF-β. Breg impairment could therefore contribute to an elevation of chronic inflammation, but the literature on that matter is scarce [339]. Decreased circulating IL-10 is a common marker of ageing (Table 3), which could be partly attributed to dampened Breg function over time. B cell ageing therefore not only accounts for decreased antibody responses to new pathogens but also contributes to the increasing prevalence of autoimmunity with age.

#### 4.1.2. T Cells

T cells express the T-cell receptor (TCR), which, like the BCR, has high specificity for individual antigens. CD4^+^ T cells can differentiate into helper (Th1, Th2, and Th17) and regulatory cells that control the activity of both B and T cells though the secretion of pro- or anti-inflammatory cytokines [340]. Upon presentation of a cognate antigen by mature DCs or B cells, CD8^+^ T cells will differentiate into cytotoxic cells capable of destroying infected or tumour cells expressing that antigen. Some T cells can end up as memory T cells by becoming quiescent [341]. The switch between effector (T_eff_) and memory (T_mem_) cells involves metabolic changes mediated in part by the kinase mTOR [342]. This is of significant interest given the finding of improved T and B cell responses to influenza vaccination and reduced incidence of respiratory tract infections in older adults treated with mTOR inhibitors [343,344,345]. While such drugs are immune-suppressive at high doses and used clinically to prevent the rejection of transplanted organs, at low doses, they are immune-protective [346].

The ageing of T cell populations starts with thymic involution in early adolescence, with increasing adiposity leading to the loss of most thymus function by adulthood; it has been suggested that involution may result from either the reduced production of haematopoietic stem cell progenitors by the bone marrow [347,348] or defects of the stromal niches of the thymus or bone marrow [349,350,351]. Notably, long-lived naked mole rats have additional thymi and decreased thymus involution associated with delayed immunosenescence [352]; hence, the retention of thymic function correlates with healthy longevity—whether this is causative is not yet known. Thymic involution is not simply an adaptive feature but instead contributes to both immunosenescence and inflammageing [353]. The resulting reduction in thymic output of functional naïve T cells with age greatly reduces the TCR repertoire [354,355,356]. Indeed, some studies suggest that over two-thirds of all T cells in older adults are CMV-reactive (i.e., most T cells only recognise cytomegalovirus), greatly reducing the antigenic repertoire (reviewed in [357]). This narrowing of the repertoire diminishes the ability to respond to new pathogens [358], leading to poor vaccine responses [129] and elevated cancer risk [359]. Unlike mice, in human adults, the thymus does not make a significant contribution to naïve T cell production [360]. With age, naïve regulatory T cells (T_reg_) also decrease, while memory regulatory T cells increase, Th1 and Th2 are maintained, and the T_reg_/T_eff_ ratio is increased [361]. Even though total numbers of pro-inflammatory Th17 decrease in ageing mice, active Th17 cells can accumulate locally in the ageing prostate and spleen, resulting in an increased Th17/T_reg_ ratio [362]. This increase in Th17 activity was also found in older humans and was the greatest contributor to their inflammatory status [363,364]. Conversely, increased IL-1β signalling and decreased IL-2 signalling both promoted Th17 polarisation in aged mice [365], which may explain how inflammageing can promote CD4^+^ differentiation skewing towards the pro-inflammatory Th17 subset. Similarly, IL-6, a major component of inflammageing, also greatly favours Th17 development to the detriment of Treg differentiation [366,367]. Failure to eliminate self-reactive T cells, potentially because of ageing defects in the medullary epithelial cells of the thymus, contributes to peripheral damage and inflammation [368,369,370]. Furthermore, the decrease in T_reg_ diversity in older adults may limit their ability to manage self-reactivity [353,371,372], leading to increasing inflammation and, together with B cell defects (see above), elevated risk of autoimmunity with age. Hence, thymic involution is likely to be an important contributor to inflammageing.

Cytotoxic T cells, which express the CD8 surface marker, are important in killing infected cells and cancer cells expressing neoantigens. IL-2 signalling by Th1 helper cells drives various CD8^+^ pathways, leading either to proliferation, differentiation, or establishment of a memory state [373]. With age, the compartment size and clonality of CD8^+^ T cells is significantly decreased [374]. Recent results have demonstrated that almost two-thirds of CD8^+^ T cells from donors aged 60 or more were senescent [375]. Additionally, aged CD8^+^ show markedly reduced cytotoxicity compared with young CD8^+^ cells, which may account for the accumulation of senescent cells in older adults, as well as elevated cancer risk, through the failure of immunosurveillance, thus indirectly participating in inflammageing. Indeed, the genetic deletion of perforin (which ablates CD8^+^ cytotoxicity) led to the increased accumulation of senescent cells and elevated chronic inflammation [37], together with tissue fibrosis and multiple age-related disorders. Notably, the removal of senescent cells alleviated many of these phenotypes [37,376], strongly supporting the idea that senescent cells drive inflammation and tissue damage.

T cell ageing and in particular senescence leads to significant changes in function [377]. While memory CD8^+^ T cells increase in number with ageing, they undergo cell cycle arrest and express the senescence marker KLRG1 (killer cell lectin-like receptor G1), and are considered non-functional [378,379]. Notably, mitochondrial dysfunction, a canonical hallmark of ageing and senescence, appears to have an important role in T cell ageing [363,380]. Due to variation in the detection of senescent T cells, their inflammatory properties are not clear in the current literature. Although senescent T cells do not seem to have increased expression of pro-inflammatory markers such as IL-6 [375], the highly differentiated CD8^+^ CD28^−^ T cells, which are the most inflammatory T cell subset, become the most represented CD8^+^ subset with age in humans [381]. In addition, age-related stiffening of the extracellular matrix has been shown to reduce T cell mobility and might lead to cell death [382]. The sestrin-driven senescence of CD8^+^ T cells has recently been reported to lead to loss of TCR expression together with the acquisition of NK features [383]. It has been suggested (but not yet tested in vivo) that this may be a positive adaptation to prevent the proliferation of T cells bearing potentially oncogenic changes, coupled with improved cell killing against chronic infections and cancers [384], but this plasticity does deplete important adaptive cells and concomitantly reinforce the pro-inflammatory innate immune system.

Overall, immunosenescence is intricately linked with inflammageing, through interactions between all components of the innate and adaptive immune systems, as well as inflammatory contributions through the SASP from senescent cells throughout the body. In the context of ageing, therefore, adaptive immune cells contribute to inflammageing through increased auto-immunity, pro-inflammatory Th17 activity, and the failure of immunosurveillance-driven removal of senescent cells, while innate immune cells drive inflammageing through myeloid skewing which increases the most inflammatory subsets, and potentially by the accumulation of cellular debris, which stimulates a pro-inflammatory response, as suggested by the garb-aging theory [5]. Senescent macrophages may be unable to clear SASP-secreting senescent cells from tissues, hence innate immune cell ageing can both directly and indirectly contribute to inflammageing. This complex interplay is summarised in Figure 5.

## 5. Strategies to Attenuate Inflammageing and Immunosenescence and Their Effects

### 5.1. Physical Exercise

Physical exercise has long been known to extend healthy lifespan. It is now becoming apparent that at least some of the health benefits arise from damping down inflammation. Lifelong endurance training is associated with reduced levels of pro-inflammatory markers [385,386,387] while interventions introducing regular endurance training for elderly people led to significant drops in inflammatory markers [388,389,390]. While intensive resistance training was associated with a brief spike in inflammatory markers such as IL-6 (potentially related to myokine function), this was then followed by an overall decrease in baseline pro-inflammatory markers in older adults [391]. A recent systematic review found that several weeks of resistance training, alone or in combination with aerobic training, was able to decrease inflammation in older adults [392], demonstrating that such exercise is an intervention that is both feasible and safe in older people including frail patients. This relationship between inflammation and physical activity goes both ways—inflammageing promotes muscle weakness and age-related sarcopenia [393,394], while physical activity is able both to substantially prevent sarcopenia and also decrease age-related inflammation [388].

Overall, long-term exercise is associated with reduced immunosenescence, with improved function of NK cells and neutrophils, reduced levels of the most pro-inflammatory monocytes subsets, increased percentage of naïve T and B cells, decreased Th17 cell polarisation, and reduced markers of senescence in T cells [99,387,395,396]. Interestingly, VO_2max_ (a marker of healthy lung function) was strongly negatively associated with senescent T cells, and adjusting for VO_2max_ removed any effect of age on the proportion of senescent and naïve T-cells [397]. A recent systematic review investigated the impact of both acute and long-term exercise in older adults, highlighting the positive effect of exercise on the removal of senescent T cells [398], which may partially account for the anti-inflammatory effect of exercise.

### 5.2. Diet

Diet may also play a role in age-related inflammation through the various inflammageing triggers discussed above. In a large cohort from Hong Kong, higher inflammatory CRP levels were associated with lower-quality diets in old men (but not in women) [399]. Similarly, a study in Spain showed a strong correlation between the self-reported intake of highly processed foods and short telomeres, which associate with senescence [400,401] and which are therefore likely to contribute to inflammageing through the SASP. While poor diet is associated with increased inflammation, the corollary is also true—a good diet can reduce inflammation. Most notably, the Mediterranean diet, rich in plant-based foods and olive oil, has been shown to have both short-term and long-term anti-inflammatory effects. For example, a sustained reduction in plasma levels of inflammatory mediators IL-1β, IL-6, IL-8, and TNF-α has been described over a period of 3 years in a free-living population with a high-risk of cardiovascular disease [402]. This reduction in inflammation seen in subjects adhering to the Mediterranean diet may occur through the lowering of oxidative stress [403]. While the links are not fully elucidated, it is likely that reducing systemic oxidative stress reduces the risk of stress-induced cellular senescence and the associated inflammatory SASP.

Dietary vitamin supplementation has also in some studies been associated with reduced inflammation. In a meta-analysis of randomised controlled trials in heart failure (HF) patients, those in the vitamin D-supplemented group had lower concentrations of TNF-α at follow-up compared with controls, but there were no differences in CRP, IL-10, or IL-6 between vitamin D and control. The authors concluded that vitamin D supplementation may have specific, but modest, effects on inflammatory markers in HF [404]. However, no such differences were detected in hypertensive patients clinically deficient for vitamin D who received supplementation [405]. Since vitamin D is synthesised in vivo on exposure to sunlight, studies can be complicated by uncontrolled biases such as the season of the year, time of the day, skin colour, or latitude, as well as interactions with other medications or intervention design [406]. The extent of suppression of inflammation by vitamin D may also depend on the drivers of such inflammation, as animals treated with diethyl nitrosamine, which drives significant levels of oxidative damage, showed significant reduction in levels of IL-1β and TNF-α when fed vitamin D in the diet [407]. Dietary vitamin D is metabolised by macrophages to its active form, and various immune cells carry vitamin D receptors, suggesting that they are responsive to it (reviewed in [408]). Notably, vitamin D can shift the balance from inflammatory Th1 and Th17 subsets of T cells towards anti-inflammatory Tregs and reduce inflammatory cytokine production by Th17 cells [409]. Hence, dietary micronutrients can directly reduce immune system-driven inflammation. Since vitamin D supplementation has been identified as one of the only dietary supplementations able to increase muscle mass in older adults [410], we speculate that it may also suppress inflammation by improving the muscle-specific production of anti-inflammatory myokines. In these ways, vitamin D may be important in reducing inflammageing.

Not only the type of food consumed, but also the quantity of food eaten, can modulate inflammation. Restriction of dietary intake without malnutrition (often known as dietary restriction, DR, or less accurately caloric restriction, or CR) increases lifespan and healthspan in many different animal models [411] and has been reported to reduce age-related inflammation [412,413]; it is also likely to trigger autophagy, which in itself may contribute to immune ‘rejuvenation’ [414]. In non-human primates, the effect of DR was found to vary according to previous dietary exposures and health status, though the overall finding was improved health and reduced inflammation [415,416,417]. While there are few research studies of DR in humans, at least one study showed a decrease in inflammation in young non-obese humans after long-term calorie restriction [418]. The mechanistic basis for DR improvements in health has not been formalised, but it is of note that the outcomes from drug inhibition of biochemical nutrient sensing pathways (e.g., mTOR inhibitors) are very similar to the DR studies in terms of better health, decreased inflammation, and extended lifespan [419]. In addition, DR may impact the gut microbiome; the microbial composition of the GI tract in young mammals includes bacterial species important in the production of anti-inflammatory short chain fatty acids [420,421,422,423,424], while microbial diversity shrinks with ageing, with a shift towards more pro-inflammatory species [425]. Pre- and pro-biotic studies have not given consistent results in terms of impacts on inflammation. However, the very recent PREDICT1 study has greatly advanced the understanding of the relationship between the gut microbiome and health, highlighting microbial biomarkers of obesity and cardiovascular disease, as well as inflammation. This study also showed that post-prandial blood glucose levels depend as much on an individual’s microbiome as on sugar intake [426]. This is important in the context of inflammation, since high circulating blood sugar levels drive the formation of advanced glycation end products (AGEs), which trigger proinflammatory RAGE signalling [57] and hence, contribute to inflammageing [58]. Hence, dietary interventions may need to be personalised to take into account microbiome, as well as food intake, in order to restrain age-related inflammation.

### 5.3. Immune Rejuvenation

Direct rejuvenation of the immune system is proving to be a fruitful avenue of research. Restoration of thymic function through the use of growth hormone (GH) in combination with metformin and DHEA (to counteract the diabetogenic action of GH) was shown in the TRIMM trial to improve overall health and reduce biological ageing, as assessed through methylation clocks [427]. The engraftment of young thymic epithelial cells into the thymus of middle-aged mice improved thymic T cell production [428], demonstrating the importance of the tissue microenvironment as well as immune cells themselves in retaining immune function. Other strategies using transplantation of induced thymic epithelial cells showed that a novel complete and functional thymus could be generated in vivo [429], which could ameliorate immune function in older patients. Thymus-targeted expression of IL-7 also rejuvenated the thymus, increased naïve T cell production, and improved immune responses to viral infection in old mice [430]. Similar results have been found when overexpressing *FOXN1* expression in the thymus of middle-aged and old mice [431,432]. Stimulating haematopoietic stem cells to proliferate by depleting aged immune cells also appears promising, since rejuvenation of the B cell lineage has been achieved by complete B cell depletion (e.g., treatment with the monoclonal anti-B cell antibody, rituximab), which triggers the expansion of progenitors; such B lineage rejuvenation was sufficient to enable treated individuals to mount a response to hepatitis B vaccination, but it did not fully restore immune system function [433,434]. The rejuvenation is thought to be a consequence of the elimination of old long-lived B cells [433], as similarly suggested in a genetic model of age-associated B cell removal [336]. Induced pluripotent stem cells derived from old HSCs were also able to restore haematopoiesis to levels seen in young mice and also avoided skewing towards myeloid lineage [435], suggesting that the epigenetic reprogramming to return to the iPSC state was sufficient to remove age-related epigenetic information. Multiple other strategies to counteract immunosenescence, such as the use of growth factor IL-7 and checkpoint inhibitors, have been investigated. Approaches may combine immunostimulatory and immunosuppressive strategies and rejuvenation of immune tissues, and have been reviewed elsewhere [436,437].

### 5.4. Interventions to Remove Senescent Cells

Treating immunosenescence by targeting drugs to the underlying ageing biology provides an important new route to ameliorate immunosenescence and inflammageing [438]. One major biological contributor to ageing and inflammageing is the accumulation of senescent cells that secrete the pro-inflammatory SASP that contributes to chronic inflammation with increasing age. Thus, the removal of senescent cells may prove useful in mitigating inflammageing by reducing the source of the SASP. Several strategies have been shown to be effective in selectively killing senescent cells, as outlined below.

Chimeric antigen receptor (CAR)-T cells can be designed to target specific antigen-expressing cellular targets. In a recent study, CAR-T cells were engineered to recognise the urokinase-type plasminogen activator receptor (uPAR) expressed on senescent cells; treatment with these CAR-T cells killed senescent cells in vitro and successfully restored tissue integrity and removed pro-inflammatory hepatic stellate senescent cells in vivo [439]. Therapeutic use of NK cells or macrophages to remove senescent cells has also been suggested [29,440].

Targeting of senescent cells with FOXO4 peptides that promote senolysis leads to better tissue integrity and apparent rejuvenation in prematurely aged mice [441]. Interestingly, such peptides were found to have the greatest effect against senescent cells with high SASP expression, and treatment led to local normalisation of IL-6 levels in the kidney tubular region of old mice [441]. Similar findings have been made with nanoparticles targeting senescent cells, wherein senescent cell removal was possibly associated with reduced SASP [442]. Moreover, the removal of senescent HSCs using the senolytic ABT263 permitted rejuvenation of aged but non-senescent HSCs and also reduced the expression of SASP factors [443]. Late-life intervention with the senolytic fisetin similarly extended lifespan and healthspan associated with a reduction in inflammatory markers in mice [444]. Clinical trials in humans with diabetic kidney disease treated with senolytic combination dasatinib with quercetin through the Translational Geroscience Network (ClinicalTrials.gov identifier: NCT02848131) have shown decreased indicators of senescence (p16 and senescence-associated β-galactosidase) in conjunction with decreased macrophages in patients; most notably, plasma SASP factors were significantly decreased. Hence, removing senescent cells to ablate the SASP is likely to be beneficial in reducing inflammageing.

Senolytic agents also appear valuable in improving immune function in ageing against infectious agents. As discussed above, viral infection in older adults exacerbates an already elevated inflammatory state. It is therefore extremely important in the context of the COVID-19 pandemic that various senolytic treatments were found to reduce inflammatory markers and greatly improve the survival of mice infected with a β-coronavirus (mouse hepatitis virus, related to SARS-CoV-2) [62,63]. The senolytic agent fisetin is now being tested in human clinical trials for prophylaxis and treatment of mild, moderate, or severe COVID-19 in older adults (ClinicalTrials.gov identifiers: NCT04537299, NCT04476953). Whether such therapies exert their beneficial effects through innate or adaptive immune cells is less clear, though antibody (i.e., B-cell mediated) responses to mouse hepatitis virus were elevated on senolytic treatment of aged animals [62].

### 5.5. Molecular Modifiers of Senescence and Inflammation

Senescent cell removal has shown significant promise in improving health and reducing inflammageing, as outlined above. However, the removal of cells, albeit senescent, has the potential to result in tissue disruption, particularly in the oldest old, and to induce functional deficits. Various approaches have been tested to suppress the damaging features of senescent cells—particularly the SASP—without eliminating them. Many of these strategies target mTOR. mTOR inhibition improves later life health and increases lifespan in mice and other experimental animals [419,445]; notably mTOR inhibitors also suppress pro-inflammatory SASP factors [15,446,447]. In human clinical studies, inhibition of mTOR successfully improved B and T cell immune function, increased vaccine response, and decreased overall infections in older adults [343,344], with stimulation of innate antiviral genes together with downregulation of pro-inflammatory factors [345], while MAPK kinase inhibitors, that overcome senescence in progeroid Werner syndrome patient cells [448], also suppress inflammation in skin fibroblasts from normally aged donors [449]. The inhibition of PI3-kinase, an upstream activator of mTOR, corrected the aberrant migration seen in old neutrophils [197]. However, a recent study reported that rapamycin improved healthspan in a model of increased systemic inflammation but did not affect inflammageing [450]. It has been suggested that differences in drug dosage may account for these inconsistencies [346]. Interestingly, statins also act on pathways regulated by mTOR, including CDC42, with the effect of improving neutrophil function and overall immune outcomes, reducing deaths from community-acquired pneumonia and even COVID-19 [438,451].

It is possible that at least some of the beneficial effects of mTOR inhibitors is exerted through the activation of autophagy, since mTOR inhibition relieves mTOR-mediated suppression of ULK-1. Indeed, spermidine, which also activates autophagy, was able to restore age-related decline in autophagy and response to vaccination and infection of CD8+ T cells in old mice [414], a finding that has now been verified in human studies examining vaccination responses in older adults [452], with human trials to improve COVID-19 vaccine responses now underway. TFEB is translated more efficiently in cells treated with spermidine [453]; since TFEB acts as a transcription factor for essential autophagy genes, the translational and autophagy effects of spermidine appear tightly linked. mTOR inhibitors similarly act on translation, as well as autophagy, relieving inhibition on eIF4 and allowing a shift from cap-dependent to cap-independent protein translation, as well as promoting the translation of oligopyrimidine tract-containing mRNAs, while spermidine improves the translation of polypeptides with polyprolines (including TFEB) through eIF5 [453]. This alters the profile of proteins synthesised by treated aged cells back to those observed in young proliferating cells.

A further pharmacological intervention to decrease inflammageing uses the very well tolerated and commonly prescribed antidiabetic drug, metformin. Metformin blocks NF-κB signalling and the downstream transcription of pro-inflammatory cytokines [454], as well as activating autophagy [363]. Epidemiological studies suggest improved health in human subjects taking metformin, even above health baselines of age-matched non-diabetic controls [455], although effects were found to be dose-dependent in ITP trials in mice [456,457]. A recent study demonstrated that treatment of CD4^+^ T cells from old donors with metformin improved autophagy and mitochondrial function, as well as limiting Th17 prevalence and its age-associated pro-inflammatory profile [363]. Metformin has also been reported to reduce autoimmunity, cytokine secretion, and NETosis [438,458,459].

Inflammasome modulators are looking promising in suppressing inflammation and improving health outcomes. A recent study showed that deacetylation of NLRP3 by SIRT2 limited age-related inflammation caused by NLRP3 activation in macrophages of old mice [460], while age-related reduction in SIRT2 expression allows NLRP3 inflammasome priming induced by mitochondrial dysfunction, impairing the regenerative capacity, and maintenance of aged HSCs [461]. Interestingly, administration of MCC950, a NLRP3 inhibitor, to aged mice improved age-related metabolic changes, including glucose tolerance and hepatic function, while inhibiting the mTOR pathway, increasing autophagy and reducing the levels of inflammation markers (active IL-1β and IL-18) [462]. NLRP3 is looking to be a very interesting target for drug inhibition to reduce inflammation without unwanted off-target effects on the wider immune system (reviewed in [463]). Sirtuin activators have also been widely tested in ageing, though at present they may be limited to diabetes [464,465,466,467]. Sirtuins require NAD^+^ to function, but NAD^+^ levels diminish markedly with age. Consistent with an important role for sirtuins in blocking biological ageing processes, supplementation with NAD^+^ (or precursors such as NMN) also appears to improve health and potentially decreases mitochondrial ROS and hence senescence and inflammation [464,468,469]. However, the wide availability of NAD^+^ as a nutraceutical dietary supplement makes placebo-controlled blinded clinical trials in humans challenging.

## 6. Conclusions

The relationship between immunosenescence and inflammageing is complex, involving the interplay between innate and adaptive arms of the immune system, together with senescent cells from non-immune system lineages, in a potentially vicious cycle. Inflammageing promotes senescence and impedes adaptive immune responses, while this impairment may lead to a greater mobilisation of innate immune cells, thereby favouring inflammageing. With the ageing of the immune system, immunosurveillance becomes less efficient, leading to failure to remove senescent cells [470]. These in turn contribute locally and systemically to inflammation. The evidence discussed above demonstrates an overall imbalance in inflammatory processes, tilting progressively towards increasing inflammation with age, with failure of anti-inflammatory molecules to counteract this. The important interconnections between inflammation, immunosenescence, frailty, and age-related disease have been highlighted further by the development of an accurate ageing clock (iAge) based on inflammatory signatures [471].

The need to improve immune health and resistance to infectious diseases in older adults has been brought into sharp focus by the COVID-19 pandemic. Herein, we have outlined several strategies to improve elderly immune function and decrease inflammation. Of these, spermidine treatment, mTOR inhibition, and the selective removal of senescent cells using senolytics are most strongly supported by model organism studies and human clinical trial data, with significant scope for immune benefit even against severe infections, such as that caused by SARS-CoV-2. It is possible that combinations of these therapies may be needed to address immunosenescence and inflammageing in the context of an ageing body burdened with accumulated senescent cells. We recommend that clinical trials of drugs for age-related diseases routinely include analysis of inflammatory mediators in order to determine whether the treatment has the added benefit of ameliorating inflammageing. We conclude that it is essential to investigate a diverse set of inflammation-related molecules to properly analyse the development of chronic inflammation with ageing; new proteomics platforms that permit simultaneous measurement of thousands of factors from very small blood or buccal swab samples should greatly facilitate such analysis and enable personalised interventions to reduce inflammation and support healthy immune function even in old age.

## Figures and Tables

**Figure 1 cells-11-00359-f001:**
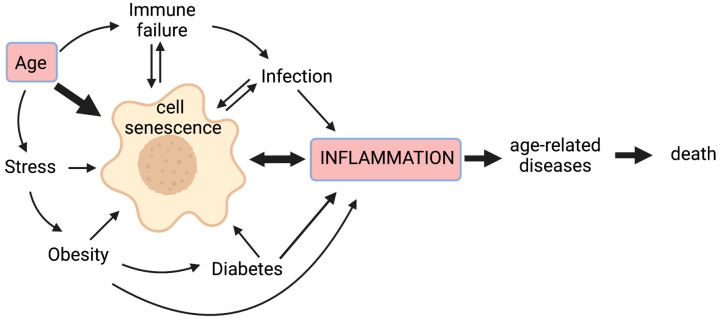
Cell senescence plays a central role in inflammageing. Cell senescence is promoted by a variety of stressors including age-related telomere attrition, DNA damage, oxidative stress, proteostatic stress, and oncogene activation. Senescence of immune cells can lead to immune failure, while age-related decreased immunity results in the poor immunological clearance of senescent cells. Obesity is associated with a pro-inflammatory state that can drive senescence, with senescent cells detected in adipose tissue; it also predisposes sufferers to diabetes, wherein altered blood sugar control may trigger senescence through glycation and metabolic stress. Inflammation is a primary response to infection, but some inflammatory signals (e.g., IL-6) can promote cell senescence. Moreover, certain viral and bacterial infections can also drive senescence. The pro-inflammatory secretome produced by senescent cells is associated with age-related diseases. Hence, senescent cells lie at the heart of a multi-faceted vicious circle leading to increased inflammation, age-related diseases, and ultimately, death.

**Figure 2 cells-11-00359-f002:**
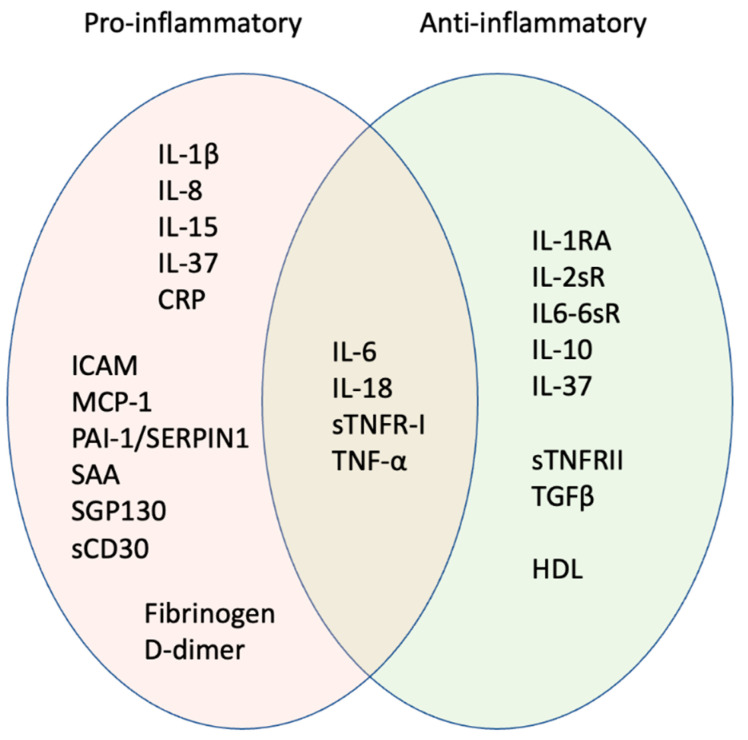
Pro-inflammatory signature of inflammageing. Orange oval shows factors most strongly associated with age-associated inflammation. Some cytokines usually considered pro-inflammatory can have anti-inflammatory activities, e.g., IL-6 produced as a cytokine through NFKB signalling pathways is pro-inflammatory, while IL-6 produced as a myokine following exercise is anti-inflammatory. Some factors may be anti-inflammatory in children and young adults but pro-inflammatory in later life; biological sex is also important in determining whether the factors act in a pro- or anti-inflammatory manner (see text for details). Hence, factors that overlap between pro- (orange) and anti-inflammatory activity (green) may be informative for the inflammatory signature.

**Figure 3 cells-11-00359-f003:**
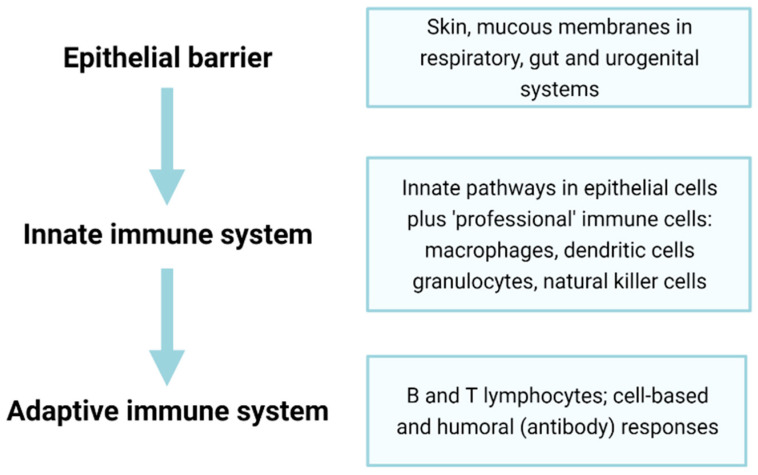
Epithelial barriers and innate and adaptive immunity constitute three major components of the immune system. Epithelial and endothelial barriers form an essential but often overlooked first line of defence in immunity. The senescence of epithelial and endothelial cells forming such barriers can lead to decreased protection (e.g., through diminished mucous secretions or ciliary activity), ‘leakiness’ through the loss of tight junctions and altered properties, including excess inflammation and tissue damage through the SASP. The second pillar of immunity, the innate immune response, is rapid-onset and, though fairly non-specific, can often be sufficient to prevent pathogens from causing disease, though senescence can lead to non-resolving inflammation. The final pillar of the adaptive immune system involves a slower-onset but highly specific response to infection; however, the senescence of these cells greatly reduces overall immune responses and may contribute to inflammageing (see text for further details).

**Figure 4 cells-11-00359-f004:**
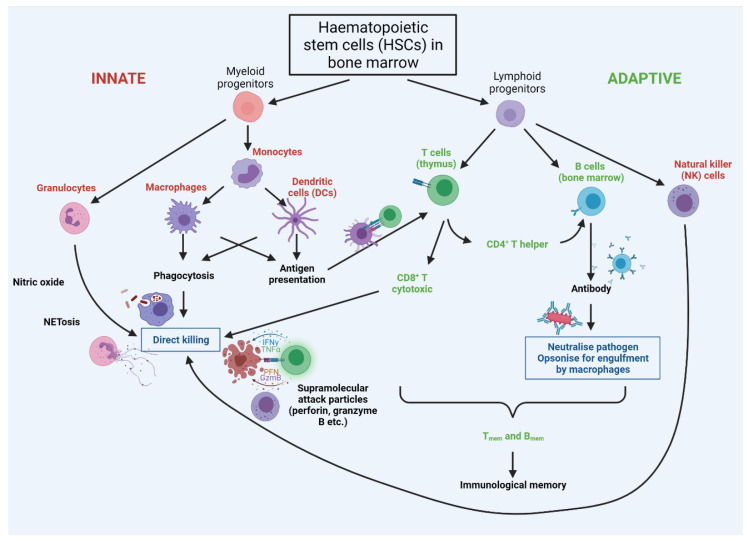
Cross-talk between cells of the innate and adaptive immune systems. Innate cells are labelled in red, with adaptive in green. Both innate and adaptive arms arise from haematopoietic stem cells (HSCs) in the bone marrow. The first line of defence is provided by the rapidly-acting PAMP/DAMP-sensing innate system, with innate cells presenting antigens to adaptive cells; this stimulates rapid proliferation of adaptive cells bearing receptors cognate for the antigen. T cells are ‘educated’ in the thymus to eliminate cells that recognise ‘self’, so that the immune system is self-tolerant. Breakage of this tolerance leads to autoimmunity, wherein the immune system attacks specific molecules or cells of the body as if they were infectious pathogens. Notably autoimmunity rises significantly with increasing age.

**Figure 5 cells-11-00359-f005:**
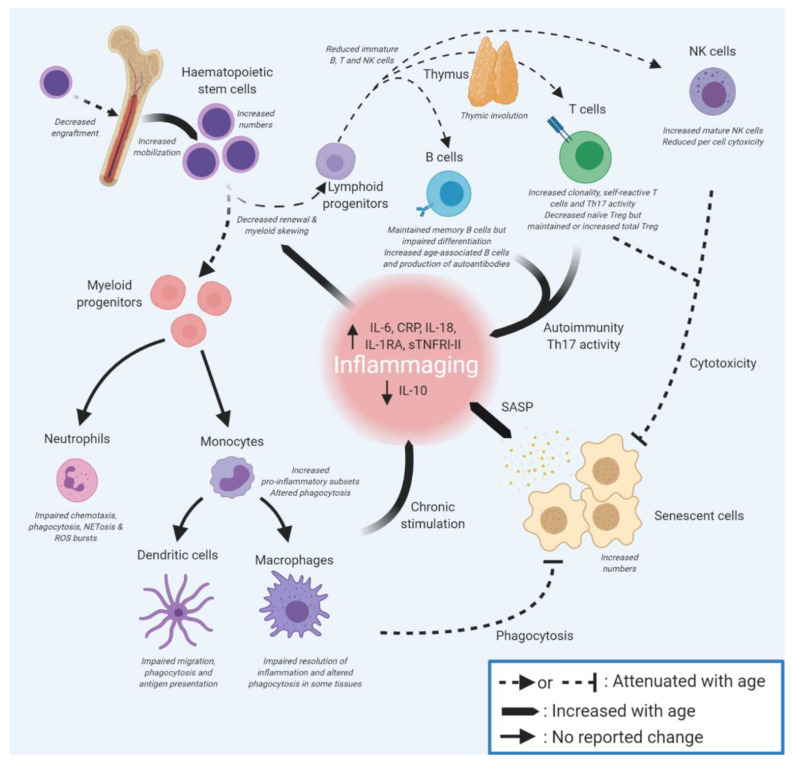
Inflammageing arises from increased cell senescence across many tissues, as well as age-related changes to the bone marrow niche and individual innate and adaptive immune cells. Together this leads to inappropriately high levels of sterile inflammation, with a number of secreted factors that may serve as a signature of inflammageing.

**Table 2 cells-11-00359-t002:** IL-18 and TNF-α as examples of factors with both pro- and anti-inflammatory activity identified in several large cohort studies.

Pro- and Anti-Inflammatory Factors		Evidence		
IL-18	TNF-α	Population Characteristics	Data Processing	Reference
PCA1—Correlated with age, mortality, morbidity and age-related diseases		1010 participants (428 men, 582 women), aged 21–96 y	Cluster analysis	[75]
Positively correlated with age	Absence of correlation with age, mortality, and morbidity. Positively correlated with some age-related diseases	General population—InCHIANTI cohort	Individual associations	
Significant independent predictor of 5- and 10-year mortality	Non-predictive of 5-year mortality	7043 participants (2995 men, 4048 women), aged 65–102 y	Fully adjusted correlation with 10-year mortality	[83]
		General population—includes CHS and InCHIANTI cohorts		
No cluster reported	“Down”-regulation—Correlated with frailty outcomes	967 participants (416 men, 551 women), aged 65 y and older	Cluster analysis	[76]
Individually correlated with worsened mobility		General population—InCHIANTI cohort		
NI	TNF-α related—Negatively correlated with knee strength, HPPB score, and grip strength; Positively correlated with 400-m walk time	1269 participants, aged 70–79 y	Cluster analysis	[77]
	Highest *R*^2^ of the TNF-α-related component for the HPPB score and knee strength	General non-frail population	Individual associations	
NI	NI	285 participants (67 men, 218 women), aged 90 y and older with a 4-year follow-up	Correlation with 4-year mortality	[93]
		General population	Fully adjusted	
Local inflammation-endothelial dysfunction	NI	320 participants (236 men, 84 women), aged 58–74 y with a 1-year follow-up	Cluster analysis	[78]
Independent predictor of 1-year recurrent coronary events		Acute coronary syndrome patients	Individual associations	
Positively correlated with age	Absence of correlation with age	1327 participants (586 men, 741 women), aged 20–102 y	Correlation with age	[80]
Reduced the size of the regression coefficient for age	NR	General population—Includes InCHIANTI cohort	Adjustment for cardiovascular risk factors and morbidities	
Proinflammation and anti-inflammation—Absence of association with 4-year death rate		580 women aged 31–85 y with a 4.7-year follow-up	Cluster analysis	[79]
		Clinically referred for coronary angiography		
NI	Negatively correlated with walking and standing balance performance	1020 participants (483 men, 537 women), aged 65–102 y	Correlation with physical performance	[81]

**Table 3 cells-11-00359-t003:** Examples of anti-inflammatory factors identified in several large population cohort studies. Note that some of these are elevated with age and poor outcomes, which may reflect the acquisition of pro-inflammatory characteristics or an unsuccessful adaptive response in an attempt to quell elevated inflammation driven by pro-inflammatory factors. (This is not an exhaustive list of anti-inflammatory factors).

IL-1RA	IL-2sR	IL-6sR	sTNFRI	sTNFRII	IL-10	TGF-β	HDL Cholesterol	Population Characteristics	Data Processing	Reference
PCA1—Correlated with age, mortality, morbidity, and age-related diseases	NI	No cluster reported	PCA1—Correlated with age, mortality, morbidity, and age-related diseases		No cluster reported	No cluster reported	NI	1010 participants (428 men, 582 women), aged 21–96 y.o	Cluster analysis	[75]
Positively correlated with age		Absence of correlation with age	Positively correlated with age, mortality, morbidity and some age-related disease	Positively correlated with age	Negatively correlated with age	Absence of correlation with age		General population—InCHIANTI cohort	Individual associations	
Significant independent predictor of 5- and 10-year mortality	NI	Non-predictive of 5-year mortality	Significant independent predictor of 5- and 10-year mortality	Non-predictive of 5-year mortality	Non-predictive of 5-year mortality	NI	NI	7043 participants (2995 men, 4048 women), aged 65–102 y.o.	Fully adjusted correlation with 10-year mortality	[83]
			Best predictor of 10-year mortality in combination with IL-6					General population—Includes CHS and InCHIANTI cohorts		
“Up”-regulation—Positively correlated with worsened mobility and frailty risk	NI	NI	NI	NI	NI	IL-1/TGF—No adverse outcome reported	NI	967 participants (416 men, 551 women), aged 65 y.o. and older	Cluster analysis	[76]
								General population—InCHIANTI cohort		
NI	TNF-α related—Negatively correlated with knee strength, HPPB score and grip strength; positively correlated with 400-m walk time				NI	NI	NI	1269 participants, aged 70–79 y.o.	Cluster analysis	[77]
	Highest *R*^2^ of the TNF-α-related component for walking speed	Highest *R*^2^ of the TNF-α-related component for grip strength	Highest *R*^2^ of the TNF-α-related component for the HPPB score and 400-m walk time					General non-frail population	Individual associations	
Positively associated with 4-year mortality	NI	NI	NI	NI	NI	NI	NI	285 participants (67 men, 218 women), aged 90 y.o. and older with a 4-year follow-up	Correlation with 4-year mortality	[93]
Strong independent predictor of 4-year mortality alone or in combination with CRP								General population	Fully adjusted	
NI	NI	NI	NI	NI	Protective or anti-inflammation—negative predictor of adverse cardiac events	NI	Protective or anti-inflammation—negative predictor of adverse cardiac events	320 participants (236 men, 84 women), aged 58–74 y.o. with a 1-year follow-up	Cluster analysis	[78]
					Significant negative predictor of 1-year adverse cardiac events		Significant negative predictor of 1-year adverse cardiac events	Acute coronary syndrome patients	Individual associations	
Positively correlated with age	NI	Positively correlated with age in women only	NI	NI	NI	Absence of correlation with age	NI	1327 participants (586 men, 741 women), aged 20–102 y.o.	Correlation with age	[80]
Removed the effect of age		Increased the size of the regression coefficient for age				NR		General population—includes InCHIANTI cohort	Adjustment for cardiovascular risk factors and morbidities	
NI	NI	NI	NI	NI	NI	Immunosuppressive—absence of association with 4-year death rate	NI	580 women aged 31–85 y.o. with a 4.7-year follow-up	Cluster analysis	[79]
								Clinically referred for coronary angiography		
Negatively correlated with global physical performance but not with hand-grip strength	NI	No significant association with physical performance	NI	NI	No significant association with physical performance	NI	NI	1020 participants (483 men, 537 women), aged 65–102 y.o.	Correlation with physical performance	[81]

## Data Availability

Not applicable.

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
