# Peer review of "Interconnections between Inflammageing and Immunosenescence during Ageing"

_cells, 2022, doi:10.3390/cells11030359_

Round 1
Reviewer 1 Report
In this review, the authors focus on ageing changes in senescence, immune system and inflammation and their interactions. They provide a nice overview of the ageing immune system and the role of inflammation, and how inflammation in ageing works. I think this is a timely and interesting article that will make a fine contribution to the literature. However, I have some comments and suggestions, as follows.
In various tables (1, 2, 3), multiple rows include data from the InCHIANTI cohort. Although the data may have been processed differently, the underlying raw data is still (at least partially) the same for all three studies. E.g. In table 1 although it is known that high IL6 levels predict mortality and the authors provide other sources, for these three rows at least it is not surprising that the results match each other given the underlying data is the same.
I think it would be more convincing to have a core set of signatures of inflammageing if the signatures were derived from a meta analysis-style approach (or even a more systematic search of pubmed) rather than the more 'cherry-picking'-like approach the authors have taken (Figure 2). The authors somewhat address this by acknowledging the lack of consistent markers analysed between different studies, but this list of markers could therefore be down to publication bias as opposed to a concrete signature of inflammageing.
It could also be interesting for the authors to perform functional enrichment of these so-called signatures using other inflammatory genes as a background - this could hint towards other commonalities between these genes (E.g. involved in similar functions or within the same pathway).
In the paragraph from lines 53 to 63 the authors outline different genes that are pro-inflammatory and associated with the SASP. A more formal analysis here would add to the manuscript - is there a statistically significant overlap between SASP genes and inflammation genes? Moreover, the authors mention in lines 44-46 that senescence is involved in various processes. Indeed, senescence can also be induced in different methods and across various cell types. The senescence phenotype, in particular the SASP, is quite heterogenous depending on method of senescence induction and cell type in which senescence is induced. It would be interesting to assess how inflammatory markers are secreted in different types of senescence/senescent cell types, as opposed to treating senescence like a homogenous process that always leads to pro-inflammatory environments. There are various databases for genes driving senescence (e.g. http://genomics.senescence.info/cells/), senescence markers (e.g. https://onlinelibrary.wiley.com/doi/full/10.1111/acel.13041 and https://pubmed.ncbi.nlm.nih.gov/28844647/), and SASP genes (e.g. http://www.saspatlas.com/), in different CS contexts that could be overlapped with pro and anti-inflammatory genes. There are also various papers looking at differential expression of genes with age across various tissues (e.g., https://pubmed.ncbi.nlm.nih.gov/33611312/) where pro- and anti-inflammatory genes could also be overlapped to strengthen the signatures of inflammageing argument.
A recent review on targeting immune dysfunction in aging, including senolytics, would be worthwhile to cite (full disclosure: I am a co-author in this paper):
https://pubmed.ncbi.nlm.nih.gov/34280555/
This recent paper on the thymus of the long-lived naked mole-rat may be worthwhile to cite as well:
https://onlinelibrary.wiley.com/doi/full/10.1111/acel.13477
Typos and minor corrections
- 'A potential' repeated twice in line 444
- 'Pay' a part instead of 'play' a part in line 267
- 'SAP' in line 714 instead of SASP
- CMV (Cytomegalovirus) in line 671 should be defined
- Line 828 says 'thorough' instead of 'through'
Overall, I think this is an interesting and timely review that will make a fine contribution to the literature, but some aspects can still be improved/clarified and ideally a more systematic gene analysis included (or at least current limitations more clearly acknowledged).
It is possible that some of my comments reflect misunderstandings of mine. If so then I would suggest that the authors use my misunderstandings as an indication that such points might be made clearer in the manuscript.
It my usual policy to reveal my identity to the authors: Joao Pedro de Magalhaes.
Author Response
We deal with the reviewer’s comments point by point below (our comments in blue).
Reviewer 1
Comments and Suggestions for Authors
In this review, the authors focus on ageing changes in senescence, immune system and inflammation and their interactions. They provide a nice overview of the ageing immune system and the role of inflammation, and how inflammation in ageing works. I think this is a timely and interesting article that will make a fine contribution to the literature. However, I have some comments and suggestions, as follows.
We thank the referee for these positive comments.
In various tables (1, 2, 3), multiple rows include data from the InCHIANTI cohort. Although the data may have been processed differently, the underlying raw data is still (at least partially) the same for all three studies. E.g. In table 1 although it is known that high IL6 levels predict mortality and the authors provide other sources, for these three rows at least it is not surprising that the results match each other given the underlying data is the same.
The point of this review was to investigate possible links between ageing and inflammation, looking at inflammageing through the lens of general cell senescence and immunosenescence. The cohort studies here are selected as they provide data from large numbers of naturally ageing individuals. We highlight where the methods for analysis of the data may vary between different studies, and we think that the confirmation of very similar results by independent teams is encouraging. Rather than starting off with the mindset that “it is known that high IL-6 levels predict mortality”, we wanted to come in with an unbiased view and then see what the actual data suggest – in fact, we discuss that IL-6 can be a marker that is both pro- and anti-inflammatory according to the cells producing it and overall physiological/pathological context.
I think it would be more convincing to have a core set of signatures of inflammageing if the signatures were derived from a meta analysis-style approach (or even a more systematic search of pubmed) rather than the more 'cherry-picking'-like approach the authors have taken (Figure 2). The authors somewhat address this by acknowledging the lack of consistent markers analysed between different studies, but this list of markers could therefore be down to publication bias as opposed to a concrete signature of inflammageing.
Our point here was not to conduct a specialist metanalysis, which would involve a major bioinformatics undertaking far outside the scope of the review, but to tie together several extremely broad fields to try to develop some overarching principles linking ageing, immunosenescence and inflammation. We have added text to the Table legends and where relevant to the main body of the text highlighting that the tables represent data from a select set of cohort studies and that they are by no means exhaustive. It is not possible for us here to overcome publication bias - a massive new multisite and potentially multinational and research programme would be required, establishing new cohorts and conducting comprehensive studies reporting levels of all factors irrespective of disease association. Here all we can do is mine the current literature and data bases in as unbiased way as possible.
It could also be interesting for the authors to perform functional enrichment of these so-called signatures using other inflammatory genes as a background - this could hint towards other commonalities between these genes (E.g. involved in similar functions or within the same pathway).
In preliminary pathway and enrichment analyses (eg KEGG), the major commonality is that the inflammatory signatures are associated with inflammation and immune functions, so we didn’t feel this added any additional information to the review.
In the paragraph from lines 53 to 63 the authors outline different genes that are pro-inflammatory and associated with the SASP. A more formal analysis here would add to the manuscript - is there a statistically significant overlap between SASP genes and inflammation genes? Moreover, the authors mention in lines 44-46 that senescence is involved in various processes. Indeed, senescence can also be induced in different methods and across various cell types. The senescence phenotype, in particular the SASP, is quite heterogenous depending on method of senescence induction and cell type in which senescence is induced. It would be interesting to assess how inflammatory markers are secreted in different types of senescence/senescent cell types, as opposed to treating senescence like a homogenous process that always leads to pro-inflammatory environments.There are various databases for genes driving senescence (e.g. http://genomics.senescence.info/cells/), senescence markers (e.g. https://onlinelibrary.wiley.com/doi/full/10.1111/acel.13041 and https://pubmed.ncbi.nlm.nih.gov/28844647/), and SASP genes (e.g. http://www.saspatlas.com/), in different CS contexts that could be overlapped with pro and anti-inflammatory genes. There are also various papers looking at differential expression of genes with age across various tissues (e.g., https://pubmed.ncbi.nlm.nih.gov/33611312/) where pro- and anti-inflammatory genes could also be overlapped to strengthen the signatures of inflammageing argument.
The referee raises an interesting point, and even though the aim of the review is not to conduct detailed bioinformatics analyses, we did attempt such functional enrichment studies as suggested by the reviewer. We extracted datasets from the databases and published papers suggested (plus some others), ran them through IPA and DAVID and carried out bioinformatics comparison analyses, but the issue is that the quality of the datasets from some of the databases is poor and there is inconsistent reporting between different papers meaning that comparisons are not really valid (eg in geneage, instead of extent of over or underexpresison, genes are scored simply as 1 if overexpressed or 0 if not - yet when we look deeper into the raw data and actual fold change, some of the changes are in the region of eg 0.02 which are not generally considered significant fold changes- this also the case in SASP atlas where changes can be quite small). We also found that some of the papers deal with RNA seq and gene over/underexpression whereas others are concerned with proteomics, and we know that concordance between RNA and protein levels can be poor. In terms of what is valid in relation to inflammageing, we suggest that the protein studies are much more important as the factors likely to drive chronic inflammation are the proteinaceous cytokines and chemokines. Conducting the analysis requested by the referee has also highlighted an issue with the published datasets and databases eg SASP atlas lacks many of the factors actually identified by the original SASP papers (even though some of those authors are also authors of the SASP atlas), so it is not particularly useful in terms of informing on inflammageing. SASP atlas also only as senescence induced by IR, ras overexpression or another drug, so it is not reflective of ‘normal’ ageing where it's more likely that replicative senescence is the major cause of the SASP.
Because of these serious issues in dataset and publication inconsistency, we felt it would detract from rather than enhance the review to include such analyses.
The referee is absolutely right in highlighting that senescence is not a homogenous state so we have written additional text to indicate that the SASP varies by cell lineage, by senescence inducer and even over time with the same cell type and inducer.
A recent review on targeting immune dysfunction in aging, including senolytics, would be worthwhile to cite (full disclosure: I am a co-author in this paper):
https://pubmed.ncbi.nlm.nih.gov/34280555/
This recent paper on the thymus of the long-lived naked mole-rat may be worthwhile to cite as well: https://onlinelibrary.wiley.com/doi/full/10.1111/acel.13477
We thank the referee for highlighting these important papers and we have now included them.
Typos and minor corrections
- 'A potential' repeated twice in line 444
- 'Pay' a part instead of 'play' a part in line 267
- 'SAP' in line 714 instead of SASP
- CMV (Cytomegalovirus) in line 671 should be defined
- Line 828 says 'thorough' instead of 'through'
Typographical errors have been corrected. We thank the referee for his attention to detail.
Overall, I think this is an interesting and timely review that will make a fine contribution to the literature, but some aspects can still be improved/clarified and ideally a more systematic gene analysis included (or at least current limitations more clearly acknowledged).
We agree that we do not set out at any time to conduct an exhaustive meta-analysis so we have clarified this in the wording of the Tables and text.
It is possible that some of my comments reflect misunderstandings of mine. If so then I would suggest that the authors use my misunderstandings as an indication that such points might be made clearer in the manuscript.
Again, the referee is right to point out that some sections may not have been as clear as we had hoped, so we have added an introductory section and clarified others.

Reviewer 2 Report
This is a timely and fairly comprehensive review of the relationship between age-related immune senescence and inflammaging. The review is well-organized and clearly written and includes well-made informative figures (except Fig 3 which is unnecessary).
The key concern is that the links between immunesenenscence and inflammaging has not been discussed in depth and only mentioned across the text and in Fig 5, this was supposed to be the focus of the review as there is an extensive list of reviews on inflammaging and immunesenescence that have been published. Furthermore , the review fails to mention that the communication is bidirectional and immunesenescence – such as increased basal secretion of pro-inflammatory cytokines ,accumulation of senescent T cells , Th17 polarisation and other features contribute towards inflammaging . This does not come across in Fig 5 as well that inflammaging could be driving immunesenescence but on the other immunesenescence has been identified as a driver of inflammaging as well.
A few minor points that need further clarification are listed below:
- The abstract is primarily focussed on inflammaging and does not mention about the connections with immunesenescence.
- The review fails to introduce the concept of inflammaging before discussing the details about how senescence contributes towards inflammaging.
- Figure 1 is lacking a figure legend
- The cross talk between the two arms of the immune system have been shown well in Figure 4; however, Fig 3 is unnecessary and doesn’t add to the narrative and can be removed.
- The B cell and ageing section mentions ABC cells and their potential role in contributing towards increased risk of autoimmunity but fails to mention about Regulatory B cells that possess anti-inflammatory properties and display numerical and functional defects as we age.
- Fig 5 and the text mentions about how inflammaging arises from senescence but the authors need to be cautious as multiple other contributing factors (microbial dysbiosis, increased adiposity, sedentary behaviour, chronic antigen exposure) have been recognised as potential contributors towards inflammaging.
- The interventions section should be possibly split into two subsections on pharmacological vs non pharmacological interventions.
- There are multiple cross-sectional and intervention studies that have reported anti-inflammaging effects and anti-immunesenescence effects (eg reversal thymic atrophy , improved neutrophil chemotaxis) which have not been mentioned .
- The section on diet should be called caloric restriction as that is the key focus and lacks mention of the anti-inflammaging effects of Mediterranean diet or Vit D supplements.
Overall , I enjoyed reading the review and look forward to reading the final version .
Author Response
We address the points raised by the reviewer point by point below (our comments are in blue text).
This is a timely and fairly comprehensive review of the relationship between age-related immune senescence and inflammaging. The review is well-organized and clearly written and includes well-made informative figures (except Fig 3 which is unnecessary).
We thank the referee for the comment, but argue that Figure 3 is necessary as the hugely important role in immunity of physical barriers (epithelia and secretions) is largely overlooked in the literature, yet barrier failure is a key aspect of immune failure that makes older adults highly susceptible to pathogens. This is seen in the skin, lung, urogenital tract, brain and the gut. Since we are discussing the failure of barrier function through the effects of cell senescence of the epithelia, we strongly argue to keep this figure in the paper.
The key concern is that the links between immunesenenscence and inflammaging has not been discussed in depth and only mentioned across the text and in Fig 5, this was supposed to be the focus of the review as there is an extensive list of reviews on inflammaging and immunesenescence that have been published. Furthermore , the review fails to mention that the communication is bidirectional and immunesenescence – such as increased basal secretion of pro-inflammatory cytokines ,accumulation of senescent T cells , Th17 polarisation and other features contribute towards inflammaging . This does not come across in Fig 5 as well that inflammaging could be driving immunesenescence but on the other immunesenescence has been identified as a driver of inflammaging as well.
The referee makes a very valid point. Although immunosenescence is often evoked in the context of inflammageing, it has not been thoroughly described in that context and many publications place macrophage activity as the key node between immunosenescence and inflammageing, but as our review suggests, the literature as it is much more complex than this. To resolve this “blurred” understanding of the involvement of immunosenescence in inflammageing, we aimed to describe all features of immunosenescence, of as many immune cell types as possible, that were relevant to inflammageing, and to include other cells such as epithelia and bone marrow cells that influence overall immune function. We found both direct roles of immunosenscence and other senescence in inflammageing, as well as indirect involvement (notably via reduced immunosurveillance of senescent cells and hence high SASP). Finally, and unexpectedly, the participation of innate immune cells in inflammageing is not as clear as it seemed to be as conflicting data exist, and they may be more relevant in the context of continuous activation by garbage accumulation. However, we did not have room to discuss all of the complexities of the garb-ageing theory. While we touch on other sources of inflammageing (infection, microbiome etc), our wish here was to focus mainly on the impact of immunosenescence and more general cell senescence on chronic age-related inflammation.
To make the aim of this review clearer to the reader, we have added a new introductory paragraph, and throughout have so we have clarified using more direct language the links that we draw between inflammageing and immunosenescence.
However, it is important that we discuss this in a balanced way as in some instances the data are not yet at a level where we can confidently make molecular links. For instance, while we point out that large number of CD8+ T cells from donors aged 60 or more are senescent , it is not clear, that senescent T cells directly participate in inflammaging. Additionally, it seems that there is often a confusion between T cell senescence and T cell exhaustion where exhaustion is reversible and exhausted cells are unable to secrete cytokines upon stimulation. (https://www.ncbi.nlm.nih.gov/pmc/articles/PMC5578132/). Recent publication even suggested that the most senescent T cells had lower IL-6 expression (https://www.ncbi.nlm.nih.gov/pmc/articles/PMC8135084/). Hence we hope that we have provided a balanced view of the state of the art without being too dogmatic in drawing links where these are not yet fully supported by data.
In term of Increased basal secretion of pro-inflammatory cytokines, we have described an overall increase in basal secretion of pro-inflammatory cytokines (but also increase and decrease of several anti-inflammatory cytokines/molecules) in the circulation, however, our current knowledge of the literature indicates that basal expression of these cytokines by the immune cells themselves is not so clear. In the case of granulocytes, conflicting data exist where some papers show an increase while others show no change in basal levels but an impaired response to stimuli.
The referee is right in pointing out the importance of Th17 so in addition to our original statement “This increase in Th17 activity was also found in older humans and was the greatest contributor to their inflammatory status ..”, we have added: “Conversely, increased IL-1β signalling and decreased IL-2 signalling both promoted Th17 polarization in aged mice, which may explain how inflammageing can promote CD4+ differentiation skewing towards the pro-inflammatory Th17 subset.”
A few minor points that need further clarification are listed below:
- The abstract is primarily focussed on inflammaging and does not mention about the connections with immunesenescence.
See above - The referee makes a very valid point so we have clarified using more direct language the links that we draw between inflammageing and immunosenescence, including in the abstract and in the new Introduction.
- The review fails to introduce the concept of inflammaging before discussing the details about how senescence contributes towards inflammaging.
Again the referee is absolutely right and we thank them for highlighting this omission which we have now corrected by providing an Introductory section.
- Figure 1 is lacking a figure legend
This has been corrected.
- The cross talk between the two arms of the immune system have been shown well in Figure 4; however, Fig 3 is unnecessary and doesn’t add to the narrative and can be removed.
See comments above as to why we are very strongly of the opinion that this figure has value.
- The B cell and ageing section mentions ABC cells and their potential role in contributing towards increased risk of autoimmunity but fails to mention about Regulatory B cells that possess anti-inflammatory properties and display numerical and functional defects as we age
This is a good point and we have now included text discussing the immunoregulatory function of Bregs. (in section 4.1.1). However, we found little to no literature regarding Breg cells in the context of ageing or inflammageing. As it is stated by an author in a recent publication investigating the effects of ageing on various regulatory cells: “Currently, it is not known whether aging directly affects the population of Bregs although Bregs are able to suppress the functions of T and B cells”. (https://www.sciencedirect.com/science/article/pii/S1568163719303198?via%3Dihub#bib0155).
- Fig 5 and the text mentions about how inflammaging arises from senescence but the authors need to be cautious as multiple other contributing factors (microbial dysbiosis, increased adiposity, sedentary behaviour, chronic antigen exposure) have been recognised as potential contributors towards inflammaging.
This is true and we do cover these important aspects of senescence and inflammation drivers in Figure 1, and in the text (for example, discussing how infection can drive senescence not only of immune cells but also eg epithelia, and how important the microbiome is in inflammation). The aim of Figure 5 is to focus on the immune system ageing in inflammageing.
- The interventions section should be possibly split into two subsections on pharmacological vs non pharmacological interventions.
We did consider this in our initial drafts but then found that the distinctions between pharmacological and non-pharmacological become blurred – at what point does dietary supplementation with a small molecule become drug treatment? Eg is spermidine a drug or a dietary supplement?
- There are multiple cross-sectional and intervention studies that have reported anti-inflammaging effects and anti-immunesenescence effects (eg reversal thymic atrophy, improved neutrophil chemotaxis) which have not been mentioned.
We have now included text on thymic reversal. We are not entirely sure what other studies the referee refers to as we do already include studies eg on neutrophil chemotaxis and improvement eg on statin treatment, improved T cells effects of spermidine treatment and overall B and T improvements on mTOR inhibition.
- The section on diet should be called caloric restriction as that is the key focus and lacks mention of the anti-inflammageing effects of Mediterranean diet or Vit D supplements.
We have now corrected these omissions and included text on the Mediterranean diet and vitamin D as suggested (as useful addition to thanks to the reference for highlighting). We have also defined DR to more accurately reflect the studies discussed.
Overall , I enjoyed reading the review and look forward to reading the final version .
We thank the reviewer for the positive and useful comments.